# Posterior Sampling via Autoregressive Generation

**Kelly W. Zhang**[*]
Department of Mathematics
Imperial College London

**Tiffany (Tianhui) Cai**[*]
Department of Statistics
Columbia University

**Hongseok Namkoong**
Decision, Risk, and Operations Division
Columbia Business School

**Daniel Russo**
Decision, Risk, and Operations Division
Columbia Business School

## Abstract

Real-world decision-making requires grappling with a perpetual lack of data as environments change; intelligent agents must comprehend uncertainty and actively gather information to resolve it. We propose a new framework for learning bandit algorithms from massive historical data, which we demonstrate in a cold-start recommendation problem. First, we use historical data to pretrain an autoregressive model to predict a sequence of repeated feedback/rewards (e.g., click responses to news articles shown to sequences of users). In learning to make accurate predictions, the model implicitly learns an informed prior based on rich action features (e.g., article headlines) and how to sharpen beliefs as more rewards are gathered (e.g., clicks as each article is recommended). At decision-time, we autoregressively sample (impute) an imagined sequence of rewards for each action, and choose the action with the largest average imputed reward. Far from a heuristic, our approach is an implementation of Thompson sampling (with a learned prior), a prominent active exploration algorithm. We prove our pretraining loss directly controls online decision-making performance, and we demonstrate our framework on a news recommendation task where we integrate end-to-end fine-tuning of a pretrained language model to process news article headline text to improve performance.

## 1   Introduction

Real-world decision-making requires grappling with a perpetual lack of data as environments change; intelligent agents must comprehend uncertainty and actively gather information to resolve it. This is especially challenging with tasks involving unstructured inputs such as text and images. This paper offers a fresh perspective, casting the problem of balancing exploration and exploitation in online decision-making as a problem of training and sampling from an autoregressive generative sequence model, an area experiencing rapid innovation [2, 23, 50].

**Problem setting.** We present our insights by deriving a novel solution to a meta-bandit problem [51, 8, 26, 3], in which an agent repeatedly encounters new tasks that require exploring to gather useful information. In real applications this meta-learning structure is common, and we illustrate our approach using a news article recommendation setting: Each day a batch of new articles is released, and upon release, the system observes each article's text but is uncertain about how engaging each article will be, as some articles may be surprise hits, or others may be less popular than expected. Models that solely rely on article text will eventually be outperformed by simple alternatives that

---

[*]Co-first authors.

Workshop on Bayesian Decision-making and Uncertainty, 38th Conference on Neural Information Processing Systems (NeurIPS 2024).

learn from repeated user feedback. This example highlights the need to use rich features (e.g., article headline) and the need to acquire further information through active exploration.

**Our Algorithm.** Our proposed solution proceeds in two phases. In the pre-training phase, the agent learns to model uncertainty by learning a simulator of user interactions using historical data on previously released articles. The simulator is an autoregressive sequence model that uses an article's attributes (e.g. headline text) to predict sequences of recommendation outcomes across users for that article. In the online decision-making phase, the agent models its uncertainty by simulating recommendation outcomes for new users with the pretrained sequence model. At each decision time, the agent uses the fitted simulator to autoregressively sample imagined/imputed recommendation outcomes for new users, conditioned on article features and on previously observed outcomes. The agent then takes the action with the greatest imputed average reward.

**Formal Connections to Thompson Sampling (TS).** Far from a heuristic, our approach is a principled implementation of TS (with a learned prior), a prominent bandit algorithm with strong guarantees [49, 45].implementations of TS, our approach never performs explicit Bayesian inference regarding latent parameters, and instead relies only on predicting and generating observable quantities. This enables standard ML tools for training. The connection between autoregressive sampling and TS rests on a link between exchangeable sequence modeling and Bayesian inference that has been known since de Finetti's seminal work [14], and has appeared in several different literatures [17, 16, 21, 15, 38, 28].

**Theoretical Guarantees and Empirical Evaluations.** We provide formal links between interactive decision-making and sequence prediction, including a novel regret bound that scales with the pre-training loss of the sequence model. Furthermore, we demonstrate that our theoretical insights bear out on a news recommendation task that incorporates a language model.

## 2  Problem formulation

**Online Decision-Making Problem.** Each online decision-making phase begins with new articles (actions) $\mathcal{A}^{\text{new}}$ being released. Each article $a \in \mathcal{A}^{\text{new}}$ is associated with attributes $Z^{(a)}$; in our experiments these represent article headlines. Even with rich article headline features $Z^{(a)}$, the system is uncertain about how engaging articles will be to readers. The system interacts sequentially with distinct users $t \in \{1, 2, \ldots, T\}$ and can adapt future recommendations based on initial user feedback. To the $t^{\text{th}}$ user, it recommends $A_t \in \mathcal{A}^{\text{new}}$, observes an outcome $Y_t$, and constructs a reward $R(Y_t) \in [0, 1]$ by applying a fixed, known function $R(\cdot)$. The vector of outcomes $Y_t$ could include a variety of user feedback like whether the user clicked or shared the recommended article.

Each action $a$ has $T$ potential outcomes $Y_{1:T}^{(a)} = (Y_1^{(a)}, ..., Y_T^{(a)})$. The observed outcome is $Y_t \leftarrow Y_t^{(A_t)}$ if article $A_t$ is recommended to the $t^{\text{th}}$ user. We assume articles are drawn independently from some fixed article distribution, i.e., $\{Z^{(a)}, Y_{1:T}^{(a)}\}$ are drawn i.i.d. across articles $a$. This assumption precludes resolving uncertainty about the effectiveness of one article by gathering feedback on a different article in the online decision-making phase. Conditioned on the article features $Z^{(a)}$, potential outcomes are drawn from a fixed and unknown distribution $p^*$:

$$Y_{1:T}^{(a)} \mid Z^{(a)} \sim p^*\big(\cdot \mid Z^{(a)}\big). \tag{1}$$

Finally, we assume $p^*$ is exchangeable, meaning that for any permutation $\sigma$ over $\{1, \ldots, T\}$, any $z$, and any outcomes $(y_1, \ldots, y_T)$,

$$p^*(y_1, \ldots, y_T \mid z) = p^*(y_{\sigma(1)}, \ldots, y_{\sigma(T)} \mid z). \tag{2}$$

Exchangeability means outcomes from recommendations made to a large subset of $m < T$ users is likely to be representative of outcomes that would have been observed among all $T$ users (Appendix J).

Our goal is to develop an adaptive algorithm $\pi$ for recommending articles that maximizes the expected average reward $\mathbb{E}_{p^*, \pi}\big[\frac{1}{T} \sum_{t=1}^{T} R(Y_t^{(A_t)})\big]$ (where the expectation is taken over draws of $\mathcal{A}^{\text{new}}$ in addition to the randomness in $p^*$ and $\pi$), or equivalently, minimizes the per-user Bayesian regret,

$$\Delta(\pi; p^*) := \mathbb{E}_{p^*, \pi}\left[ \max_{a \in \mathcal{A}^{\text{new}}} \left\{ \frac{1}{T} \sum_{t=1}^{T} R(Y_t^{(a)}) \right\} - \frac{1}{T} \sum_{t=1}^{T} R(Y_t^{(A_t)}) \right]. \tag{3}$$

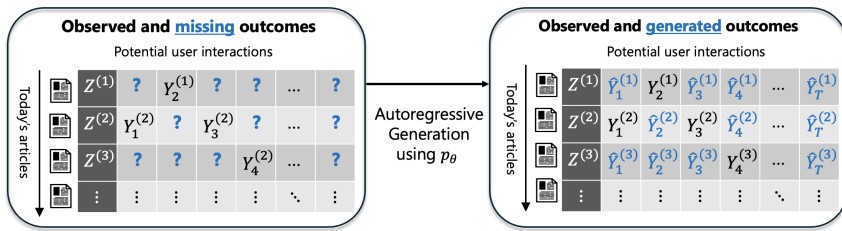

Figure 1: **Missing data viewpoint.** We view uncertainty about unobserved outcomes as the source of uncertainty for decision-making, avoiding explicit reference to latent parameters/variables. Calibrated generation (imputation) of missing outcomes enables uncertainty quantification and good decisions.

In (3), we calculate the gap in reward relative to a baseline that always recommends the action with best performance in the (finite) population.

**Pretraining Phase.** The goal of the pre-training phase is to learn a good active exploration algorithm to deploy in the online decision-making phase. We have access to a historical dataset $\mathcal{D}^{\text{hist}} := \left\{ Z^{(a)}, Y_{1:n}^{(a)} : a \in \mathcal{A}^{\text{hist}} \right\}$, with action attributes $Z^{(a)}$ and observed outcomes $Y_{1:n}^{(a)}$ from previous articles (actions) $a \in \mathcal{A}^{\text{hist}}$, for some $n \leq T$. We assume this dataset is drawn from the same data generating distribution as in the online decision-making phase: Across $a \in \mathcal{A}^{\text{hist}}$, $Z^{(a)} \overset{i.i.d}{\sim} P_Z$ and $Y_{1:n}^{(a)}$ is a completely random subset of size $n$ of $Y_{1:T}^{(a)}$, where $Y_{1:T}^{(a)} \mid Z^{(a)} \sim p^*(\cdot \mid Z^{(a)})$.

# 3 Posterior Sampling via Autoregressive Generation

This work considers *unobserved outcome data* as the source of a decision-maker's uncertainty (Figure 1): for a given article, we only have responses from some users, and there is residual uncertainty in how future users would respond. Inspired by this viewpoint, our method proceeds in two steps:

**Phase 1: Pretraining an Autoregressive Model.** We train an autoregressive sequence model $p_\theta$, with parameter $\theta \in \Theta$, that can predict missing outcomes, conditioned on article (action) attributes, and limited previously observed outcomes. This enables us to generate hypothetical completions of the potential outcome table in Figure 1. Formally, this model specifies a probability $p_\theta(Y_t^{(a)} \mid Z^{(a)}, Y_{1:t-1}^{(a)})$ of observing outcome $Y_t^{(a)}$ in the next interaction given article attributes $Z^{(a)}$ and previous outcomes $Y_{1:t-1}^{(a)}$. We minimize the following loss on the historical dataset $\mathcal{D}^{\text{hist}}$:

$$\ell(p_\theta; \mathcal{D}^{\text{hist}}) = - \sum_{a \in \mathcal{A}^{\text{hist}}} \log p_\theta(Y_{1:n}^{(a)} \mid Z^{(a)}) = - \sum_{a \in \mathcal{A}^{\text{hist}}} \sum_{t=1}^{n} \log p_\theta(Y_t^{(a)} \mid Z^{(a)}, Y_{1:t-1}^{(a)}). \quad (4)$$

Our approach to pre-training an approximate exchangeable sequence model can also be thought of as empirical Bayes (Appendix C). Our approach also mirrors recent work on neural processes [20, 25, 35, 28] and prior-data fitted networks [33]. Our main contribution is linking this pretrained sequence model to online decision-making, which we present next.

**Phase 2: Online Decision-Making via Autoregressive Generation.** After a sequence model $p_\theta$ is trained on historical data, it is deployed and used for decision-making. No additional training of $p_\theta$ is needed. At each decision time, our algorithm uses $p_\theta$ to autoregressively generate imputed values of missing outcomes for each candidate action $a \in \mathcal{A}^{\text{new}}$, as seen in Figure 2. At decision time $t$, let $\mathcal{T}_{\text{miss}}^{(a)}$ denote indices of the users $\tau \in [1:T]$ for which article/action $a$ has not been recommended so far. The algorithm samples (imputes) outcomes $\hat{Y}_\tau^{(a)}$ for each $\tau \in \mathcal{T}_{\text{miss}}^{(a)}$ conditional on article attributes $Z^{(a)}$, as well as previously observed and generated outcomes for article $a$. It then uses both the observed and generated outcomes to compute an imputed mean reward for action $a$:

$$\hat{\mu}_t^{(a)} \leftarrow \frac{1}{T} \left\{ \sum_{\tau \in [1:T] \text{ s.t. } \tau \notin \mathcal{T}_{\text{miss}}^{(a)}} R\big(Y_\tau^{(a)}\big) + \sum_{\tau \in \mathcal{T}_{\text{miss}}^{(a)}} R\big(\hat{Y}_\tau^{(a)}\big) \right\}. \quad (5)$$

Finally, the algorithm selects action $A_t = \text{argmax}_{a \in \mathcal{A}^{\text{new}}} \left\{ \hat{\mu}_t^{(a)} \right\}$. Then the real outcome $Y_t^{(A_t)}$ is observed. The process is repeated at the next decision time. See Algorithm 2 for further details.

Through this process, actions that are optimal under some likely generation of the missing outcomes according to $p_\theta$, have a chance of a being selected. Once no plausible sample of missing outcomes

could result in an action being optimal, it is essentially written off. Good performance of the algorithm relies on the model $p_\theta$ matching the data generating process closely.

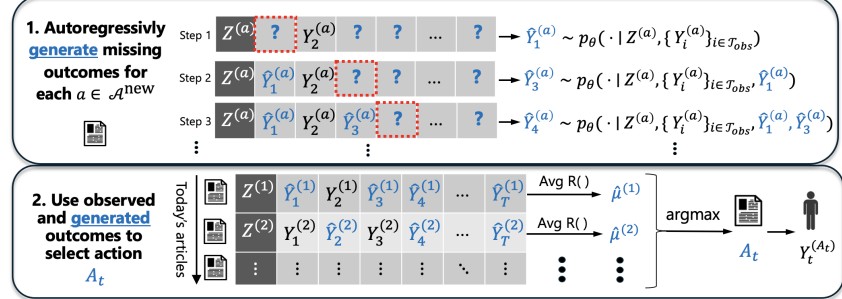

Figure 2: **Posterior Sampling via Autoregressive Generation (PS-AR).**

### 3.1 Interpreting our Decision-Making Algorithm as Thompson Sampling (TS)

We now formalize how the generated/imputed outcomes faithfully represent uncertainty and that PS-AR is akin to Thompson (posterior) sampling, which selects actions proportionally to the probability that actions are optimal. Lemma 1 shows that the imputed mean $\hat{\mu}_t^{(a)}$ from PS-AR is a posterior sample of the mean reward $\mu^{(a)}$, and the action $A_t$ selected by PS-AR is a posterior sample of the optimal action $A^* := \arg\max_{a \in \mathcal{A}^{\text{new}}} \{\mu^{(a)}\}$ where $:= \frac{1}{T}\sum_{t=1}^{T} R(Y_t^{(a)})$. For simplicity, Lemma 1 is stated under the assumption that PS-AR uses the optimal sequence model $p^*$ (see E.2 for result for *approximate* models $p_\theta$). Let $\mathcal{H}_t := (\{Z^{(a)}\}_{a \in \mathcal{A}^{\text{new}}}, A_1, Y_1, \ldots, A_t, Y_t)$ denote history up to time $t$.

**Lemma 1.** *Under Algorithm 2 applied with $p_\theta = p^*$, for all $a \in \mathcal{A}^{\text{new}}$, with probability 1,*

$$\mathbb{P}\big(\hat{\mu}_t^{(a)} = \cdot \mid \mathcal{H}_{t-1}\big) = \mathbb{P}\big(\mu^{(a)} = \cdot \mid \mathcal{H}_{t-1}\big) \quad \text{and} \quad \mathbb{P}\big(A_t = a \mid \mathcal{H}_{t-1}\big) = \mathbb{P}_{p_\theta}\big(A^* = a \mid \mathcal{H}_{t-1}\big).$$

Corollary 1 formalizes how expected loss of the learned sequence model $p_\theta$ controls the regret of PS-AR, reducing a sequential decision-making problem to loss minimization. Proofs in Appendix E.

**Corollary 1.** *For PS-AR (Algorithm 2) applied with $p_\theta$, which we denote as $\pi_{\text{PS-AR}}(p_\theta)$,*

$$\Delta\big(\pi_{\text{PS-AR}}(p_\theta); p^*\big) \leq \underbrace{\sqrt{\frac{|\mathcal{A}^{\text{new}}|\log(|\mathcal{A}^{\text{new}}|)}{2T}}}_{\text{Regret bound for Thompson sampling}} + \underbrace{\sqrt{\frac{|\mathcal{A}^{\text{new}}|}{2}\big\{\ell_T(p_\theta) - \ell_T(p^*)\big\}}}_{\text{Penalty for sub-optimal prediction}}.$$

**Advantages of the Autoregressive Approach.** Since our approach focuses on predicting missing outcomes, the learned model only needs to model *observable* quantities, and can be learned via loss minimization (4). In contrast, a more standard perspective on TS requires specifying a model for latent variables and performing explicit Bayesian inference; for large scale problems this often involves simplifying modeling assumptions, expensive MCMC, or heuristic posterior approximations.

### 3.2 News Recommendation Experiments

We build a news recommendation task using the MIcrosoft News Dataset (MIND) [53] where we demonstrate how PS-AR easily integrates with pretrained language models. Rewards are binary (click/no-click). We consider three types of sequence models. FLEXIBLE NN (TEXT) and BETA-BERNOULLI NN (TEXT) are neural network models that incorporate article headlines using DistilBERT [46]. FLEXIBLE NN (CATEGORY) uses only category information (e.g. "Sports"). See Appendix F.1 for synthetic experiments, and Appendix G for experiment details.

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

# A  Posterior Sampling via Autoregressive Generation (PS-AR) Algorithm

We present pseudo-code for the two phases of our method: pre-training (Algorithm 1) and online decision-making (Algorithm 2).

---

### Algorithm 1: Pretraining an autoregressive model

**Require:** Training data $\mathcal{D}^{\text{hist}}$, model class $\{p_\theta\}_{\theta \in \Theta}$, batch size $b$

1: **while** not converged **do**
2:      Sample a minibatch $\mathcal{D} = \{Z^{(a)}, Y_{1:n}^{(a)}\}_{a \in \mathcal{A}}$ where $\mathcal{A} \subset \mathcal{A}^{\text{hist}}$, $|\mathcal{A}| = b$
3:      For each $a \in \mathcal{A}$, sample outcomes with replacement:

$$\tilde{Y}_1^{(a)}, \tilde{Y}_2^{(a)}, \dots \tilde{Y}_T^{(a)} \mid \{Z^{(a)}, Y_{1:n}^{(a)}\} \overset{i.i.d.}{\sim} \frac{1}{n} \sum_{i=1}^n \delta_{Y_i^{(a)}}$$

         Above, $\frac{1}{n} \sum_{i=1}^n \delta_{Y_i^{(a)}}$ denotes the empirical distribution of $Y_{1:n}^{(a)}$

4:      Define bootstrap-resampled minibatch $\tilde{\mathcal{D}} \leftarrow \{Z^{(a)}, \tilde{Y}_{1:T}^{(a)}\}_{a \in \mathcal{A}}$
5:      Compute loss $\ell(p_\theta; \tilde{\mathcal{D}})$ as defined in (4)
6:      Backpropagate and take a gradient step to update $\theta$
7: **end while**
8: **return** $p_\theta$

---

**Practical Considerations for the Pre-Training Step (Algorithm 1)**

- In Algorithm 1 line 3, instead of bootstrapping sequences of length $T$, for practical purposes we sometimes bootstrap samples sequences of length $T_{\text{train}} < T$ if training to length $T$ is very computationally expensive (we do this for our news recommendation experiments).

- While we have been assuming that $\mathcal{D}^{\text{hist}}$ has sequences all of the same length $n$, in practice, this may not always be the case. Let $n^{(a)}$ refer to the number of observations for article $a \in \mathcal{A}^{\text{hist}}$. In this case, we can easily replace $n$ with $n^{(a)}$ in line 3 of Algorithm (2) (we do this for our news recommendation experiments).

---

### Algorithm 2: Posterior Sampling via Autoregressive Generation (PS-AR)

**Require:** Autoregressive generative model $p_\theta$, evaluation actions $\mathcal{A}^{\text{new}}$ with attributes $\{Z^{(a)}\}_{a \in \mathcal{A}^{\text{new}}}$

1: Initialize observed user indices list $\mathcal{T}_{\text{obs}}^{(a)} \leftarrow [\,]$ for each $a \in \mathcal{A}^{\text{new}}$
2: **for** $t = 1, \dots, T$ **do**
3:      **for** $a \in \mathcal{A}^{\text{new}}$ **do**
4:          Initialize list of generated user indices $\mathcal{T}_{\text{gen}}^{(a)} \leftarrow [\,]$
5:          **for** $\tau \in [1 : T]$ such that $\tau \notin \mathcal{T}_{\text{obs}}^{(a)}$ **do**
6:              Autoregressively sample hypothetical outcomes for missing values

$$\hat{Y}_\tau^{(a)} \sim p_\theta\big( \cdot \mid Z^{(a)}, \, \big(Y_i : i \in \mathcal{T}_{\text{obs}}^{(a)}\big), \, \big(\hat{Y}_i^{(a)} : i \in \mathcal{T}_{\text{gen}}^{(a)}\big)\big)$$

7:              Update generated set $\mathcal{T}_{\text{gen}}^{(a)} \leftarrow \text{append}\big(\mathcal{T}_{\text{gen}}^{(a)}, \tau\big)$
8:          **end for**
9:          Form imputed average reward using observed and generated outcomes

$$\hat{\mu}_t^{(a)} \leftarrow \frac{1}{T} \bigg\{ \sum_{\tau \in \mathcal{T}_{\text{obs}}^{(a)}} R\big(Y_\tau\big) + \sum_{\tau \in \mathcal{T}_{\text{gen}}^{(a)}} R\big(\hat{Y}_\tau^{(a)}\big) \bigg\}$$

10:     **end for**
11:     Select action $A_t \leftarrow \text{argmax}_{a \in \mathcal{A}^{\text{new}}} \big\{\hat{\mu}_t^{(a)}\big\}$, breaking ties deterministically.
12:     Update observed user lists $\mathcal{T}_{\text{obs}}^{(A_t)} \leftarrow \text{append}\big(\mathcal{T}_{\text{obs}}^{(A_t)}, t\big)$
13:     Observe outcome $Y_t$ from action $A_t$.
14: **end for**

---

**Remark 1** (Including Observed Rewards in the Average). *In Algorithm 2 we average over both observed rewards and imputed values of unobserved rewards. Including observed rewards helps sharpen theoretical understanding: it lets us say the algorithm is exactly a finite population (i.e. a very large group of $T$ users) version of Thompson sampling that is used in the online learning literature [6, 5]. However, it has little practical bearing on performance if $T$ is large. See Fig 4 in Appendix A.*

**Practical Considerations for the Online Step (Algorithm 2)**

- For practical purposes, we may generate $m$ outcomes rather than imputing all unobserved outcomes in $[1 : T]$ to save computation. Specifically, replace lines 4-9 in Algorithm 2 with Algorithm 3 below.

---

Algorithm 3: Truncated Autoregressive Posterior Generation

---

**Require:** Autoregressive generative model $p_\theta$, timestep $t$, generation length $m$, action attribute $Z^{(a)}$, previously observed outcomes $(Y_i : i \in \mathcal{T}_{\text{obs}}^{(a)})$ for that action

1: **for** $t' = 1, 2, \ldots m$ **do**
2:     Autoregressively sample hypothetical outcomes for missing values

$$\hat{Y}_{t'}^{(a)} \sim p_\theta\big( \cdot \mid Z^{(a)}, \big(Y_i : i \in \mathcal{T}_{\text{obs}}^{(a)}\big), \hat{Y}_{1:t'-1}^{(a)}\big)$$

3: **end for**
4: Form imputed average reward using $m$ generated outcomes

$$\hat{\mu}_t^{(a)} \leftarrow \sum_{t'=1}^{m} R\big(\hat{Y}_{t'}^{(a)}\big)$$

---

## A.1 Empirical Comparisons of PS-AR Variants

**Examining Full Imputation vs Truncated Generation (Figure 4)**   We empirically compare PS-AR (Algorithm 2), i.e., "full imputation", with the computationally cheaper version of PS-AR that truncates generation to a maximum of length of $m$ (Algorithm 2 with lines 4-9 replaced with Algorithm 3). We find that both versions of PS-AR perform well in practice and that the original PS-AR (Algorithm 2) and the $m$-truncated version (with $m = 500$) have similar performance.

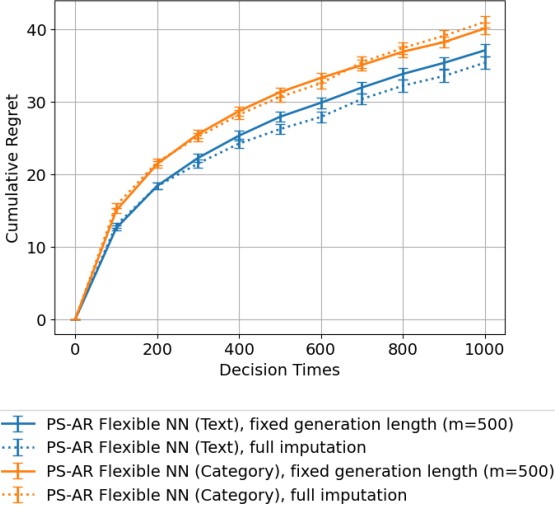

Figure 4: **Full imputation vs. truncated generation of future rewards.** Error bars are $\pm 1$ s.e. averaged over 500 runs.

Specifically in Figure 4 we compare both versions of PS-AR on a news recommendation setting. In our experimental setup, use two versions of the pretrained autoregressive sequence model $p_\theta$: FLEXIBLE NN (TEXT) and FLEXIBLE NN (CATEGORY) (see Appendix G.3 for more details). We run both versions of PS-AR with each of these two $p_\theta$ models. We use $T = 1000$, $|\mathcal{A}^{\text{new}}| = 10$, and the truncated version of PS-AR uses $m = 500$. We follow the procedure described Appendix G.3 in forming the regret plots: we run 500 repetitions of each bandit algorithm and in each repetition we draw a new set of 10 actions/articles from the validation set to represent a "new task". Regret is calculated with respect to $\mu^{(a)}$ in Equation (6) below:

$$\mu^{(a)} := \frac{1}{T} \sum_{t=1}^{T} R(Y_t^{(a)}) \qquad \text{and} \qquad A^* := \operatorname{argmax}_{a \in \mathcal{A}^{\text{new}}} \left\{ \mu^{(a)} \right\}. \qquad (6)$$

As discussed in Remark 1, the version of the algorithm that performs full imputation averages over both previously observed rewards and hypothetical samples of unobserved rewards when computing the imputed mean (5). The $m$-truncated version averages solely over generated rewards. This difference does not have a practically significant impact on performance in Figure 4.

**Examining Truncating Generation Length (Figure 5)** We examine the performance of our PS-AR algorithm for different generation truncation lengths $m$ (Algorithm 2 with lines 4-9 replaced with Algorithm 3). Throughout all our previous experiments we use $m = 500$. In Figure 5, we examine the impact of varying $m$ on the regret of the PS-AR with the FLEXIBLE NN (TEXT) sequence model in the news recommendation setting. We follow the procedure described Appendix G.3 in forming the regret plots: we run 500 repetitions of each bandit algorithm and in each repetition we draw a new set of 10 actions/articles from the validation set to represent a "new task". We find that that increasing $m$ reduces the regret of the algorithm; however, when $m$ is sufficiently large, the benefit of increasing $m$ further is negligible.

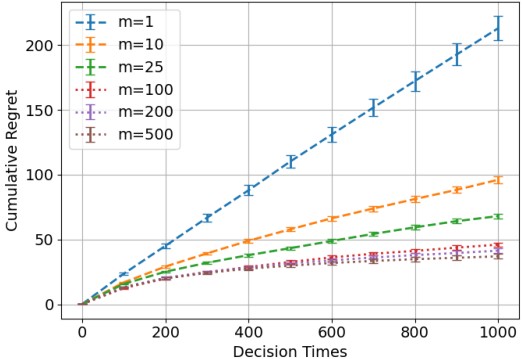

Figure 5: **Examining Truncating Generation Length ($|\mathcal{A}^{\text{new}}| = 10$).** Error bars are $\pm 1$ s.e. averaged over 500 runs.

## B  Finite vs infinite population formulations and Thompson sampling variants

This section discusses the intimate connections between (large) finite-population formulations that were discussed in the main body of the paper and infinite-population formulations that are more common in the Bayesian bandit literature. We do this in the special case of the mixture model of Example 1.

We emphasize that **from our perspective, the main advantages or disadvantages of the finite population view are conceptual.** In terms of advantages: (1) the definitions do not require any explicit assumptions around mixture modeling or latent variables. and (2) The finite nature of the problem lets us visualize the procedure as in Figure 2, without abstract reference to limits across infinite sequences.

## B.1 Review of Thompson sampling in infinite populations, with mixture models.

Thompson sampling is most often defined for a mixture model as in Example 1. Following that example, we consider in the subsection the canonical example of exchangeable sequences: a mixture model wherein the outcomes are i.i.d conditioned on a latent variable $U^{(a)}$. That is, $p^*(Y_1^{(a)}, \ldots, Y_T^{(a)} \mid Z^{(a)}) = \int \prod_{t=1}^{T} P(Y_t^{(a)} \mid Z^{(a)}, U^{(a)} = u) P(U^{(a)} = u) du$. The unknown latent variable represents the decision-maker's uncertainty about an action's performance.

The literature typically defines the "true arm means" as

$$\mu_\infty^{(a)} = \int R(y) P(y \mid Z^{(a)}, U^{(a)}) dy.$$

The subscript highlights that this has the interpretation of a long-run average reward across an infinite population of users (or infinite set of rounds). By the law of large numbers (applied conditional on $(Z^{(a)}, U^{(a)})$), one has

$$\mu_\infty^{(a)} = \lim_{T \to \infty} \frac{1}{T} \sum_{t=1}^{T} R(Y_t^{(a)}).$$

The true best arm is defined as

$$A_\infty^* \in \operatorname{argmax}_{a \in \mathcal{A}^{\text{new}}} \mu_\infty^{(a)}$$

Randomness in the latent parameters $(U^{(a)})$ means $\mu_\infty^{(a)}$ and $A_\infty^*$ are random variables whose realizations are uncertain even given the history $\mathcal{H}_{t-1}$. Thompson sampling selects an action by probability matching on $A_\infty^*$, defined by the property

$$\mathbb{P}(A_t = a \mid \mathcal{H}_{t-1}) = \mathbb{P}(A_\infty^* = a \mid \mathcal{H}_{t-1}) \quad \text{for all } a \in \mathcal{A}^{\text{new}}. \tag{7}$$

Per-period Bayesian regret over $T$ periods is defined as

$$\mathbb{E}\left[ \frac{1}{T} \sum_{t=1}^{T} \left( R\left(Y_t^{(A^*)}\right) - R\left(Y_t^{(A_t)}\right) \right) \right] \tag{8}$$

## B.2 Thompson sampling in finite populations

One can define the true mean of a finite population as

$$\mu_T^{(a)} = \frac{1}{T} \sum_{t=1}^{T} R(Y_t^{(a)}).$$

The true best arm for this finite population is defined as

$$A_T^* \in \operatorname{argmax}_{a \in \mathcal{A}^{\text{new}}} \mu_T^{(a)}$$

As in Lemma 1, Thompson sampling selects an action by probability matching on the (finite-population) optimal action $A_T^*$, defined by the property

$$\mathbb{P}(A_t = a \mid \mathcal{H}_{t-1}) = \mathbb{P}(A_T^* = a \mid \mathcal{H}_{t-1}) \quad \text{for all } a \in \mathcal{A}^{\text{new}}. \tag{9}$$

Per-period Bayesian regret over $T$ periods is defined as

$$\mathbb{E}\left[ \frac{1}{T} \sum_{t=1}^{T} \left( R\left(Y_t^{(A_T^*)}\right) - R\left(Y_t^{(A_t)}\right) \right) \right] \tag{10}$$

It is not hard to show that (10) is a more stringent notion of regret than in (8), since $\frac{1}{T} \sum_{t=1}^{T} R\left(Y_t^{(A_T^*)}\right) \geq \frac{1}{T} \sum_{t=1}^{T} R\left(Y_t^{(A_\infty^*)}\right)$ by definition of $A_T^*$. Both definitions are widely used, with the more stringent finite-population version being more common in the adversarial bandit literature; see [27].

## B.3 The gap between finite and infinite population formulations is small

We analyze the gap between the two formulations in the case of a mixture model. Let $\mathcal{U}^{\text{new}} = \{U^{(a)} : a \in \mathcal{A}^{\text{new}}\}$ and recall $\mathcal{Z}^{\text{new}} = \{Z^{(a)} : a \in \mathcal{A}^{\text{new}}\}$. By a sub-Gaussian maximal inequality

$$\mathbb{E}\left[\max_{a \in \mathcal{A}^{\text{new}}} \left|\mu_\infty^{(a)} - \mu_T^{(a)}\right|\right] = \mathbb{E}\left[\mathbb{E}\left[\max_{a \in \mathcal{A}^{\text{new}}} \left|\mu_\infty^{(a)} - \mu_T^{(a)}\right| \mid \mathcal{Z}^{\text{new}}, \mathcal{U}^{\text{new}}\right]\right] \leq \sqrt{\frac{2\log(|\mathcal{A}^{\text{new}}|)}{T}},$$

To justify the last inequality, note that since the function $R$ takes values in $[0, 1]$, $R(Y_t^{(a)}) - \mu_\infty^{(a)}$ is subgaussian with variance proxy 1, conditional on $\mathcal{Z}^{\text{new}}, \mathcal{U}^{\text{new}}$ (by Hoeffing's Lemma). Since it is the average of independent sub-Gaussian random variables, $\mu_\infty^{(a)} - \mu_T^{(a)}$ is subgaussian with variance proxy $\frac{1}{T}$, conditional on $\mathcal{Z}^{\text{new}}, \mathcal{U}^{\text{new}}$. The last step follows then from applying the subgaussian maximal inequality, conditional on $\mathcal{Z}^{\text{new}}, \mathcal{U}^{\text{new}}$.

It follows easily that the infinite population optimum $A_\infty^*$ is near optimal for finite populations:

$$0 \leq \mathbb{E}\left[\max_{a \in \mathcal{A}^{\text{new}}} \mu_T^{(a)} - \mu_T^{(A_\infty^*)}\right] \leq 2\sqrt{\frac{2\log(|\mathcal{A}^{\text{new}}|)}{T}}.$$

Analogously, the finite population optimum is near-optimal in infinite populations:

$$0 \leq \mathbb{E}\left[\max_{a \in \mathcal{A}^{\text{new}}} \mu_\infty^{(a)} - \mu_\infty^{(A_T^*)}\right] \leq 2\sqrt{\frac{2\log(|\mathcal{A}^{\text{new}}|)}{T}}.$$

Supported by this theory, we do not focus on the distinction between $A_T^*$ and $A_\infty^*$ in our developments.

## B.4 Similar Insights in Empirical Results

Some empirical insight can also be gleaned from Figure 4 in Appndix B. The implementation that performs full imputation can be interpreted as Thompson sampling for a finite population. As discussed in Remark 1, averages over both past observed rewards and samples of hypothetical unobserved rewards when sampling hypthetical population means.

The implementation that performs forward generation of fixed-length $m$ does not include past observed rewards in the average. For very large $m$, it is a direct approximation to infinite-horizon Thompson sampling. We can see in Figure 4 the these implementations have very similar performance.

## C Interpreting our Training Loss

Define the The expected analogue of the training loss (4) (and averaged over the draw of news articles) is

$$\ell_n(p_\theta) := \mathbb{E}\left[-\sum_{t=1}^n \log p_\theta\left(Y_t^{(a)} \mid Z^{(a)}, Y_{1:t-1}^{(a)}\right)\right]. \tag{11}$$

The next lemma is a standard result connecting the excess expected loss of a sequence model $p_\theta$ to its KL divergence from the true sequence model $p^*$. To (nearly) minimize loss, $p_\theta$ the learner needs to closely approximate the true sequence model.

**Lemma 2.** *For any sequence model $p_\theta$,*

$$\ell_n(p_\theta) = \ell_n(p^*) + \mathbb{E}_{Z^{(a)} \sim P_Z}\left[D_{\text{KL}}\left(p^*\left(Y_1^{(a)}, \ldots, Y_n^{(a)} \mid Z^{(a)}\right) \,\Big\|\, p_\theta\left(Y_1^{(a)}, \ldots, Y_n^{(a)} \mid Z^{(a)}\right)\right)\right].$$

Following the lemma, we provide an example that notes connections to ideas in Bayesian statistics.

**Example 1** (Exchangeability and mixture models)**.** *The canonical example of exchangeable sequences is mixture models, where the outcomes are i.i.d conditioned on a latent variable $U^{(a)}$. That is, $p^*(Y_1^{(a)}, \ldots, Y_T^{(a)} \mid Z^{(a)}) = \int \prod_{t=1}^T P(Y_t^{(a)} \mid Z^{(a)}, U^{(a)} = u)P(U^{(a)} = u)du$. The unknown latent variable represents the decision-maker's uncertainty about an action's performance.*

**Example 2** (Empirical Bayes)**.** *Under the mixture model from Example 1, $p^*$ is called a posterior predictive distribution in Bayesian statistics. Consider the case where $p_\theta$ is a posterior predictive*

*induced by prior hyper-parameters $\theta$. For ease of exposition, imagine a setting with no $Z$'s and a conjugate Bayesian model where $\mu^{(a)} \sim \text{Beta}(\alpha, \beta)$ and $Y_1^{(a)}, Y_2^{(a)}, \cdots \mid \mu^{(a)} \overset{i.i.d.}{\sim} \text{Bernoulli}(\mu^{(a)})$. By Bayes rule, the posterior predictive distribution is*

$$p_\theta\big(Y_{t+1}^{(a)} = 1 \mid Y_{1:t}^{(a)}\big) = \frac{\alpha + \sum_{i=1}^{t} Y_i^{(a)}}{\alpha + \beta + t} \quad \text{where} \quad \theta = (\alpha, \beta). \tag{12}$$

*For this choice of $p_\theta$, our training criterion (4) is equivalent to that used in Empirical Bayes (Type-II maximum likelihood) to fit prior distributions to observed data [34, 7, 36]. We empirically observe that training on our sequence loss can recover the true Bayesian prior (Appendix G.5). Our pretraining procedure can be viewed as learning an approximate posterior predictive by gradient descent.*

We discuss connections between our pretraining procedure and empirical Bayes (12) in Appendix G.5.

## D  Extension to the Contextual Setting

In this section we discuss a preliminary approach to extend our algorithm to the setting with context features. In the news recommendation setting, the context features would represent user features.

**Data Generating Process.**    In this setting, article features $Z^{(a)}$ are drawn independently from $P_Z$ over $a \in \mathcal{A}^{\text{new}}$. Independently of that, user contexts $X_t$ are discrete and drawn i.i.d. from an unknown distribution $P_X$, i.e., $X_1, X_2, \ldots, X_T \overset{i.i.d.}{\sim} P_X$. Then,

$$Y_t^{(a)} \mid Z^{(a)}, (X_{t'}, Y_{t'}^{(a)})_{t'=t}^{t-1}, X_t \sim p^*\big(\cdot \mid Z^{(a)}, (X_{t'}, Y_{t'})_{t'=t}^{t-1}, X_t\big). \tag{13}$$

Moreover, $p^*$ is such that for any $z$ and any permutation $\sigma$ over $\{1, \ldots, T\}$,

$$\big(X_1, Y_1^{(a)}\big), \ldots, \big(X_T, Y_T^{(a)}\big) \mid (Z^{(a)} = z) \overset{D}{=} \big(X_{\sigma(1)}, Y_{\sigma(1)}^{(a)}\big), \ldots, \big(X_{\sigma(T)}, Y_{\sigma(T)}^{(a)}\big) \mid (Z^{(a)} = z),$$

where above we use $\overset{D}{=}$ to denote equality in distribution.

The historical dataset follows the same data generating process.    For shorthand, we use $(X, Y^{(a)})_{1:n} := \big((X_1, Y_1^{(a)}), \ldots, (X_n, Y_n^{(a)})\big)$ to denote sequences of tuples. We denote the training set $\mathcal{D}^{\text{hist}} = \big\{ Z^{(a)}, (X, Y^{(a)})_{1:n} \big\}$ (for some $n \leq T$). For each $a \in \mathcal{A}^{\text{hist}}$, we assume $(X, Y^{(a)})_{1:n}$ is a completely at random subset of the tuples $(X, Y^{(a)})_{1:T}$ where $X_1, X_2, \ldots, X_T \overset{i.i.d.}{\sim} P_X$ and $Y_{1:T}^{(a)}$ are sampled according to (13).

**Phase 1: Pretraining an auto-regressive model.**    We train a sequence model analogously to Algorithm 1, however replace the training loss (4) with the following loss:

$$\ell(p_\theta; \mathcal{D}^{\text{hist}}) = \sum_{a \in \mathcal{A}^{\text{hist}}} \left[ -\sum_{t=1}^{n} \log p_\theta\left(Y_t^{(a)} \mid Z^{(a)}, (X, Y^{(a)})_{1:t-1}, X_t^{(a)}\right) \right]. \tag{14}$$

In the contextual case, transformers are a natural choice for the sequence model architecture for $p_\theta$.

---

Algorithm 4: Pretraining an autoregressive model with Context

---

**Require:** Training data $\mathcal{D}^{\text{hist}}$, model class $\{p_\theta\}_{\theta \in \Theta}$, batch size $b$

1: **while** not converged **do**
2:       Sample a minibatch $\mathcal{D} = \{Z^{(a)}, (X, Y^{(a)})_{1:n}\}_{a \in \mathcal{A}}$ where $\mathcal{A} \subset \mathcal{A}^{\text{hist}}$, $|\mathcal{A}| = b$
3:       For each $a \in \mathcal{A}$, sample outcomes with replacement:

$$(\tilde{X}_1, \tilde{Y}_1^{(a)}), (\tilde{X}_2, \tilde{Y}_2^{(a)}), \ldots (\tilde{X}_T, \tilde{Y}_T^{(a)}) \mid \{Z^{(a)}, (X, Y^{(a)})_{1:n}\} \overset{i.i.d.}{\sim} \frac{1}{n} \sum_{i=1}^{n} \delta_{(X_i, Y_i^{(a)})}$$

        Above, $\frac{1}{n}\sum_{i=1}^{n} \delta_{(X_i, Y_i^{(a)})}$ denotes the empirical distribution of $(X, Y^{(a)})_{1:n}$.

4:       Define bootstrap-resampled minibatch $\tilde{\mathcal{D}} \leftarrow \{Z^{(a)}, (\tilde{X}, \tilde{Y}^{(a)})_{1:T}\}_{a \in \mathcal{A}}$
5:       Compute loss $\ell(p_\theta; \tilde{\mathcal{D}})$ using (14)
6:       Backpropagate and take a gradient step to update $\theta$
7: **end while**
8: **return** $p_\theta$

---

**Phase 2: Online decision-making via autoregressive generation**   Online decision-making with the pre-trained $p_\theta$ sequence model can be made using Algorithm 5 below. Similar to the version without context, it generates missing outcomes.

---

Algorithm 5: Posterior Sampling via Autoregressive Generation (PS-AR) with Context

---

**Require:** Autoregressive generative model $p_\theta$, evaluation actions $\mathcal{A}^{\text{new}}$ with attributes $Z^{(a)}$

1: Initialize observed user indices list $\mathcal{T}_{\text{obs}}^{(a)} \leftarrow [\,]$ for each $a \in \mathcal{A}^{\text{new}}$
2: **for** $t = 1, \ldots, T$ **do**
3:       Observe user context $X_t$ and set $x \leftarrow X_t$
4:       **for** $a \in \mathcal{A}^{\text{new}}$ **do**
5:          Initializes list of generated user indices $\mathcal{T}_{\text{gen}}^{(a)} \leftarrow [\,]$
6:          **for** $\tau \in [1 : T]$ such that $\tau \notin \mathcal{T}_{\text{obs}}^{(a)}$ **do**
7:             Autoregressively sample hypothetical outcomes for missing values

$$\hat{Y}_\tau^{(a)} \sim p_\theta\big( \cdot \mid Z^{(a)}, \big(X_i^{(a)}, Y_i^{(a)} : i \in \mathcal{T}_{\text{obs}}^{(a)}\big), \big(x, \hat{Y}_i^{(a)} : i \in \mathcal{T}_{\text{gen}}^{(a)}\big), X_\tau = x\big)$$

8:             Update generated set $\mathcal{T}_{\text{gen}}^{(a)} \leftarrow \text{append}\big(\mathcal{T}_{\text{gen}}^{(a)}, \tau\big)$
9:          **end for**
10:         Form hypothetical average reward using observed and generated (imputed) outcomes

$$\hat{\mu}_t^{(a)} \leftarrow \frac{1}{|\mathcal{T}_{\text{gen}}^{(a)}| + |\sum_{\tau \in \mathcal{T}_{\text{obs}}^{(a)}} \mathbb{1}_{X_\tau = x}|} \left\{ \sum_{\tau \in \mathcal{T}_{\text{obs}}^{(a)}} R(Y_\tau^{(a)}) \mathbb{1}_{X_\tau = x} + \sum_{\tau \in \mathcal{T}_{\text{gen}}^{(a)}} R(\hat{Y}_\tau^{(a)}) \right\}$$

11:       **end for**
12:       Select action $A_t \leftarrow \text{argmax}_{a \in \mathcal{A}^{\text{new}}} \big\{\hat{\mu}_t^{(a)}\big\}$, breaking ties deterministically.
13:       Update observed user lists $\mathcal{T}_{\text{obs}}^{(A_t)} \leftarrow \text{append}\big(\mathcal{T}_{\text{obs}}^{(A_t)}, t\big)$
14:       Observe outcome $Y_t$ from action $A_t$.
15: **end for**

---

# E  Theoretical Results

Corollary 1 relies on Theorem 1, which is a result that may be of independent interest that bounds the regret of *any* policy $\pi$ in terms of the regret of $\pi$ on the simulated environment and the gap in sequence loss between the simulated environment and the true data generating environment, $\ell_T(p_\theta) - \ell_T(p^*)$. Below we use $\Delta(\pi; p_\theta)$ to denote the regret of $\pi$ when the potential outcomes $Y_{1:T}^{(a)}$ are generated autoregressively from $p_\theta$ given $Z^{(a)}$ for each $a \in \mathcal{A}^{\text{new}}$ (see Appendix E.3 for more details).

**Theorem 1.** *For **any** policy $\pi$,* $\underbrace{\Delta(\pi; p^*)}_{\text{Deployment regret}} \leq \underbrace{\Delta(\pi; p_\theta)}_{\text{Regret under simulator}} + \underbrace{\sqrt{(|\mathcal{A}^{\text{new}}|/2)\{\ell_T(p_\theta) - \ell_T(p^*)\}}}_{\text{Penalty for sub-optimal simulator}}.$

Theorem 1, a novel result, states that the regret achieved by any algorithm $\pi$ under the fitted environment simulator $p_\theta$ is close to the regret $\pi$ will achieve when deployed in the true environment $p^*$, so long as the loss achieved by $p_\theta$ is close to that of $p^*$. Theorem 1 thus characterizes the regret of any algorithm, including TS, that uses a misspecified prior. To see this, pick $\pi$ to be Thompson sampling with a misspecified prior and pick $p_\theta$ to be the data generating distribution under the misspecified prior; then $\Delta(\pi; p_\theta)$ will have the typical regret bound for Thompson sampling and the second term on the RHS of (1) characterizes the penalty for having a misspecified prior. Our result can be thought of in some ways as a generalization of [47], which only applied to a "k-shot" version of Thompson sampling.

See proofs in the rest of this section.

## E.1  Proof of Lemma 2

*Proof.* By the definition of the expected loss in (11), and the chain rule of KL divergence:

$$\ell_n(p_\theta) - \ell_n(p^*)$$
$$= \mathbb{E}\left[-\sum_{t=1}^n \log p_\theta\big(Y_t^{(a)} \mid Z^{(a)}, Y_{1:t-1}^{(a)}\big)\right] - \mathbb{E}\left[-\sum_{t=1}^n \log p^*\big(Y_t^{(a)} \mid Z^{(a)}, Y_{1:t-1}^{(a)}\big)\right]$$
$$= \text{KL}\big(\mathbb{P}_{p^*}(Y_1^{(a)}, \ldots, Y_n^{(a)} \mid Z^{(a)}) \,\|\, \mathbb{P}_{p_\theta}(Y_1^{(a)}, \ldots, Y_n^{(a)} \mid Z^{(a)})\big)$$
$$= \mathbb{E}_{Z^{(a)} \sim P_Z}\left[\text{KL}\big(\mathbb{P}_{p^*}(Y_1^{(a)}, \ldots, Y_n^{(a)} \mid Z^{(a)}) \,\|\, \mathbb{P}_{p_\theta}(Y_1^{(a)}, \ldots, Y_n^{(a)} \mid Z^{(a)})\big)\right].$$

The final equality is the definition of the KL divergence between conditional distributions. $\square$

## E.2  Posterior sampling interpretation: proof of Lemma 1

*Proof.* At decision time $t$, suppose in the history $\mathcal{H}_{t-1}$ a particular action $a \in \mathcal{A}^{\text{new}}$ has been shown to users $\mathcal{T}_{\text{obs}}^{(a)} \subseteq \{1, 2, \ldots, t-1\}$. For the function $f(\{y_i\}_{i=1}^T) = T^{-1}\sum_{i=1}^T R(y_i)$, one has

$$\mu^{(a)} = f\big(\{Y_i^{(a)}\}_{i=1}^T\big) \qquad \text{and} \qquad \hat{\mu}_i^{(a)} = f\big(\{Y_i^{(a)} : i \in \mathcal{T}_{\text{obs}}^{(a)}\} \cup \{\hat{Y}_i^{(a)} : i \in \mathcal{T}_{\text{gen}}^{(a)}\}\big)$$

where $\{\hat{Y}_i^{(a)} : i \in \mathcal{T}_{\text{gen}}^{(a)}\}$ are drawn according to Algorithm 2 applied with sequence model $p_\theta = p^*$. The result that

$$\mathbb{P}\big(\hat{\mu}_t^{(a)} = \cdot \mid \mathcal{H}_{t-1}\big) = \mathbb{P}_{p_\theta}\big(\mu^{(a)} = \cdot \mid \mathcal{H}_{t-1}\big)$$

follows immediately since $\{Y_i^{(a)} : i \in \mathcal{T}_{\text{obs}}^{(a)}\}$ are non-random conditioned on $\mathcal{H}_{i-1}$ and $\mathbb{P}\big((\hat{Y}_t^{(a)} : i \in \mathcal{T}_{\text{gen}}^{(a)}) = \cdot \mid \mathcal{H}_{t-1}\big) = \mathbb{P}\big((Y_t^{(a)} : t \in \mathcal{T}_{\text{gen}}^{(a)}) = \cdot \mid \mathcal{H}_{t-1}\big)$ with probability 1. The proof of the analogous result for $A^*$ is identical. $\square$

## E.3  Interpreting the data-generating process corresponding to a mispecfied non-exchangeable sequence model

Our model assumes the true sequence model $p^*$ is exchangeable. To derive our theory, we want to view posterior sampling by auto-regressive sampling (Algorithm 2) as a proper implementation of Thompson sampling, with approximation coming solely from the incorrect use of a sequence model $p_\theta$. To make this rigorous, we need to correctly interpret the data-generating process under when $p_\theta$ is not exchangeable. Under what data-generating process would the algorithmic generation procedure in Figure 2 still be correct? The definition below turns out to be the right one.

**Definition 1** (Outcome revelation order under non-exchangeable sequence models). *A (possibly non-exhangeable) sequence model $p_\theta$ introduces an alternative data-generating process as follows. First, independently for each arm $a \in \mathcal{A}^{\text{new}}$, nature samples arm features $Z^{(a)} \sim P_Z$; then it samples $Y_{1:T}^{(a)} \mid Z^{(a)} \sim p_\theta(\cdot \mid Z^{(a)})$. If arm $A_t = a$ is selected at time $t$ and this is the $k^{\text{th}}$ time that arm is chosen, then $Y_t \leftarrow Y_k^{(A_t)}$.*

We note that this data generating process is simply specifying the order in which outcomes from the sequence model are revealed to the decision-maker. Namely, we view $Y_t^{(a)}$ as the potential outcome of the $t^{\text{th}}$ play of arm $a$ whereas the main body of the paper views $Y_t^{(a)}$ as the potential outcome for the $t^{\text{th}}$ user/period. Under an exchangeable sequence models, order is irrelevant and the two data generating processes are mathematically equivalent. **Note that this alternative data generating processes is a proof technique, and not part of the model of the problem.**

### E.4 Posterior sampling interpretation: generalization of Lemma 1

The next result generalizes Lemma 1. The proof is the same, but we use the indexing convention in Definition 2.

**Lemma 3.** *Under Algorithm 2 applied with $p_\theta$,*

$$\mathbb{P}\big(\hat{\mu}_t^{(a)} = \cdot \mid \mathcal{H}_{t-1}\big) = \mathbb{P}_{p_\theta}\big(\mu^{(a)} = \cdot \mid \mathcal{H}_{t-1}\big) \tag{15}$$

*and for all $a \in \mathcal{A}^{\text{new}}$,*

$$\mathbb{P}\left(A_t = a \mid \mathcal{H}_{t-1}\right) = \mathbb{P}_{p_\theta}\left(A^* = a \mid \mathcal{H}_{t-1}\right). \tag{16}$$

*Proof.* We use the notation of Definition 1. Let $N_t^{(a)} = \sum_{i=1}^t \mathbb{1}(A_t = a)$ denote the number of times arm $a$ was played upto and including period $t$. Then, the observation at time $t$ is

$$Y_t \leftarrow Y_{N_t^{(A_t)}}^{(A_t)}.$$

For the function $f(\{y_i\}_{i=1}^T) = T^{-1} \sum_{i=1}^T R(y_i)$, one has

$$\mu^{(a)} = f\big(\{Y_i^{(a)}\}_{i=1}^T\big) \qquad \text{and} \qquad \hat{\mu}_t^{(a)} = f\big(\{Y_1^{(a)}, \ldots, Y_{N_t^{(a)}-1}^{(a)}\} \cup \{Y_{N_t^{(a)}}^{(a)}, \ldots, Y_T^{(a)}\}\big)$$

where $(Y_{N_t^{(a)}}^{(a)}, \ldots, Y_T^{(a)}) \sim p_\theta\big(\cdot \mid Z^{(a)}, Y_{1:N_t^{(a)}-1}^{(a)}\big)$ represent the generated outcomes drawn according to Algorithm 2 applied with sequence model $p_\theta$. Property (15) follows immediately since $\{Y_1^{(a)}, \ldots, Y_{N_t^{(a)}-1}^{(a)}\}$ are non-random conditioned on the history $\mathcal{H}_{t-1}$ and $\mathbb{P}\big((Y_{N_t^{(a)}}^{(a)}, \ldots, Y_T^{(a)}) = \cdot \mid \mathcal{H}_{t-1}\big) = \mathbb{P}\big((Y_{N_t^{(a)}}^{(a)}, \ldots, Y_T^{(a)}) = \cdot \mid \mathcal{H}_{t-1}\big)$ with probability 1. The proof of (16) is identical. $\square$

### E.5 Proof of Theorem 1

We continue to use the indexing convention in Definition 1 of Subsection E.3. Formally, any policy $\pi$ can be expressed a function that maps a history $\mathcal{H}_{t-1}$ and an exogenous random seed $\xi$ to an action as

$$A_t = \pi(\mathcal{H}_{t-1}, \xi). \tag{17}$$

The random seed allows for algorithmic randomness in action selection and is assumed to be independent of the draws of article features and potential outcomes $(Z^{(a)}, Y_{1:T}^{(a)})_{a \in \mathcal{A}^{\text{new}}}$.

The essence of the proof is to recognize that one could write a simulator that first randomly drew the environment "sample path" $(Z^{(a)}, Y_{1:T}^{(a)})_{a \in \mathcal{A}^{\text{new}}}$ and the algorithm seed $\xi$, and then implemented a completely deterministic sequence of operations to calculate the regret an algorithm incurs with that sample path and seed. Mathematically, the simulator is a function, (written as $g(\cdot)$ in the proof). We can view mis-specification of the sequence model as mis-specifying the distribution of the sample path draws used in the the simulator. We use information-theoretic tools to bound the impact this distributional change on the inputs to the simulator can have on the distribution of outputs of the simulator (e.g. regret).

*Proof.* Since this proof requires analyzes regret under the mis-specified and possibly non-exchangeable model $p_\theta$, we must be precise about the order in which potentail outcomes are revealed. See Definition 2 for discussion of our indexing convention.

This proof will show that for any policy $\pi$,

$$\Delta(\pi; p^*) \leq \Delta(\pi; p_\theta) + \sqrt{\frac{|\mathcal{A}^{\text{new}}|}{2} \{\ell_T(p_\theta) - \ell_T(p^*)\}}.$$

Note for any policy $\pi$, by the triangle inequality,

$$\Delta(\pi; p^*) \leq |\Delta(\pi; p^*) - \Delta(\pi; p_\theta)| + |\Delta(\pi; p_\theta)|$$

The remainder of the proof will focus on bounding the first term above. Let $S^{\text{new}} := \{Z^{(a)}, Y_{1:T}^{(a)} : a \in \mathcal{A}^{\text{new}}\}$ denote a draw of all article features and potential outcomes.

The absolute difference in regret can be written as

$$|\Delta(\pi; p_\theta) - \Delta(\pi; p^*)|$$
$$= \left| \mathbb{E}_{p^*, \pi} \left[ \max_{a \in \mathcal{A}^{\text{new}}} \left\{ \frac{1}{T} \sum_{t=1}^{T} R(Y_t^{(a)}) \right\} - \frac{1}{T} \sum_{t=1}^{T} R(Y_t) \right] - \mathbb{E}_{p_\theta, \pi} \left[ \max_{a \in \mathcal{A}^{\text{new}}} \left\{ \frac{1}{T} \sum_{t=1}^{T} R(Y_t^{(a)}) \right\} - \frac{1}{T} \sum_{t=1}^{T} R(Y_t) \right] \right|$$
$$= \left| \mathbb{E}_{p^*} \left[ g(S^{\text{new}}, \xi) \right] - \mathbb{E}_{p_\theta} \left[ g(S^{\text{new}}, \xi) \right] \right|,$$

where $g$ is is a function that determines the algorithm's regret as a function of the potential outcomes and the external seed $\xi$ that used to induce randomness in action actions. That is, $g(\{Z^{(a)}, Y_{1:T}^{(a)}\}_{a \in \mathcal{A}^{\text{new}}}, \xi) := \frac{1}{T} \sum_{t=1}^{T} \left\{ R(Y_t^{(A^*)}) - R(Y_t^{(A_t)}) \right\}$.

This implies

$$|\Delta(\pi; p_\theta) - \Delta(\pi; p^*)|$$
$$\underbrace{\leq}_{(i)} \sup_{f : \|f\|_\infty \leq 1} \left\{ \mathbb{E}_{p^*} \left[ f(S^{\text{new}}, \xi) \right] - \mathbb{E}_{p_\theta} \left[ f(S^{\text{new}}, \xi) \right] \right\}$$
$$\underbrace{\leq}_{(ii)} \sqrt{\frac{1}{2} \text{KL}(\mathbb{P}_{p^*}(S^{\text{new}}, \xi) \| \mathbb{P}_{p_\theta}(S^{\text{new}}, \xi))}$$
$$\underbrace{=}_{(iii)} \sqrt{\frac{1}{2} \underbrace{\text{KL}(\mathbb{P}_{p^*}(\xi) \| \mathbb{P}_{p_\theta}(\xi))}_{=0} + \frac{1}{2} \text{KL}(\mathbb{P}_{p^*}(S^{\text{new}} \mid \xi) \| \mathbb{P}_{p_\theta}(S^{\text{new}} \mid \xi))}$$
$$\underbrace{=}_{(iv)} \sqrt{\frac{1}{2} \cdot \text{KL}(\mathbb{P}_{p^*}(S^{\text{new}}) \| \mathbb{P}_{p_\theta}(S^{\text{new}}))}$$
$$\underbrace{=}_{(v)} \sqrt{\frac{|\mathcal{A}^{\text{new}}|}{2} \cdot \text{KL}(\mathbb{P}_{p^*}(Z^{(a)}, Y_{1:T}^{(a)}) \| \mathbb{P}_{p_\theta}(Z^{(a)}, Y_{1:T}^{(a)}))}$$
$$\underbrace{=}_{(vi)} \sqrt{\frac{|\mathcal{A}^{\text{new}}|}{2} \cdot \sqrt{\underbrace{\text{KL}(\mathbb{P}_{p^*}(Z^{(a)}) \| \mathbb{P}_{p_\theta}(Z^{(a)}))}_{=0} + \text{KL}(\mathbb{P}_{p^*}(Y_{1:T}^{(a)} \mid Z^{(a)}) \| \mathbb{P}_{p_\theta}(Y_{1:T}^{(a)} \mid Z^{(a)}))}}$$
$$\underbrace{=}_{(vii)} \sqrt{\frac{|\mathcal{A}^{\text{new}}|}{2} \{\ell_T(p_\theta) - \ell_T(p^*)\}}$$

- (i) holds because $g$ is a function that takes values in $[-1, 1]$, so $\|g\|_\infty \leq 1$.

- (ii) holds by Fact 9 in [44] (which uses Pinsker's inequality).

- (iii) holds the chain rule for Kullback Liebler Divergence.

- (iv) holds because $\xi$ is and $S^{\text{new}}$ are independent.

- (v) and (vi) hold again because the $(Z^{(a)}, Y^{(a)}_{1:T})$ are i.i.d. across $a \in \mathcal{A}^{\text{new}}$ and the chain rule for Kullback Liebler Divergence.

- (viii) holds by Lemma 2.

$\square$

## E.6 Proof of Corollary 1

This proof is largely review of an information-theoretic analysis of Thompson sampling due to [44]. It was observed by [6, 5] that this analysis applied without modification to analyze regret with respect to the best fixed action ($A^*$) even in nonstationary environments (e.g. non-exchangeable models $p_\theta$ as in Definition 1.)

*Proof.* We continue to use the indexing convention in Definition 1. This proof will show that for any sequence model $p_\theta$,

$$\Delta\big(\pi_{\text{PS-AR}}(p_\theta); p^*\big) \leq \sqrt{\frac{|\mathcal{A}^{\text{new}}|\log(|\mathcal{A}^{\text{new}}|)}{2T}} + \sqrt{\frac{|\mathcal{A}^{\text{new}}|}{2}\big\{\ell_T(p_\theta) - \ell_T(p^*)\big\}}.$$

By Theorem 1,

$$\Delta\big(\pi_{\text{PS-AR}}(p_\theta); p^*\big) \leq \Delta\big(\pi_{\text{PS-AR}}(p_\theta); p_\theta\big) + \sqrt{\frac{|\mathcal{A}^{\text{new}}|}{2}\big\{\ell_T(p_\theta) - \ell_T(p^*)\big\}}.$$

We bound $\Delta\big(\pi_{\text{PS-AR}}(p_\theta); p_\theta\big)$ by combining the probability matching result of Lemma 1 with Thompson sampling regret bound techniques from Russo and Van Roy [44]. Define the regret for the $t^{\text{th}}$ action as

$$\Delta_t := R\left(Y^{(A^*)}_{N_t^{(A^*)}}\right) - R\left(Y^{(A_t)}_{N_t^{(A_t)}}\right).$$

The notation $Y^{(A^*)}_{N_t^{(A^*)}}$ is discussed in Definition 1 and refers to the outcome of playing arm $A^*$ for the $N_t^{(A^*)}$-th time. With this definition,

$$\Delta\big(\pi_{\text{PS-AR}}(p_\theta); p_\theta\big) = \mathbb{E}_{p_\theta, \pi_{\text{PS-AR}}(p_\theta)}\left[\frac{1}{T}\sum_{t=1}^{T}\Delta_t\right].$$

By the proof of Proposition 1 of [44] (which is general and applies to all algorithms),

$$\mathbb{E}_{p_\theta, \pi_{\text{PS-AR}}(p_\theta)}\left[\frac{1}{T}\sum_{t=1}^{T}\Delta_t \,\middle|\, \mathcal{Z}^{\text{new}} = z\right] \leq \sqrt{\frac{H_{p_\theta}\big(A^* \mid \mathcal{Z}^{\text{new}} = z\big)\cdot\Gamma}{T}}$$

$$\leq \sqrt{\frac{\log(|\mathcal{A}^{\text{new}}|)\cdot\Gamma}{T}}$$

where $H_{p_\theta}(A^* \mid \mathcal{Z}^{\text{new}} = z) \leq \log(|\mathcal{A}^{\text{new}}|)$ refers to the conditional Shannon entropy of $A^*$ given $\mathcal{Z}^{\text{new}} := (Z^{(a)})_{a \in \mathcal{A}^{\text{new}}} = z$ under the data generating process defined by $p_\theta$, and $\Gamma$ is a constant upper bound on the "information ratio" such that

$$\Gamma \geq \max_{t \in [1:T]}\left\{\frac{\big(\mathbb{E}_t[\Delta_t]\big)^2}{I_t\big(A^*; (A_t, Y_t)\,|\,\big)}\right\} \quad \text{w.p. 1.}$$

Above we use $\mathbb{E}_t[\cdot] = \mathbb{E}_t[\cdot \mid \mathcal{H}_{t-1}]$ to denote that expectations are conditioned on the history and $I_t\big(A^*; (A_t, Y_t)\big)$ to denote the mutual information between $A^*$ and $(A_t, Y_t)$ conditional evaluated under a base measure that conditions on $\mathcal{H}_{t-1}$. (Recall that the history also includes the information in $\mathcal{Z}^{\text{new}}$).

The proof of Proposition 5 of [44] shows that one can choose $\Gamma \leq |\mathcal{A}^{\text{new}}|/2$ w.p. 1. As observed in [6, 5], this proof relies only on the probability matching property in Lemma 3 and hence applies in our setting.

Combining our results implies

$$\mathbb{E}_{p_\theta, \pi_{\text{PS-AR}}(p_\theta)}\left[\frac{1}{T}\sum_{t=1}^{T}\Delta_t\,\middle|\,\mathcal{Z}^{\text{new}} = z\right] \leq \sqrt{\frac{\log(|\mathcal{A}^{\text{new}}|)\cdot|\mathcal{A}^{\text{new}}|}{2T}},$$

so the result follows by the law of iterated expectations. □

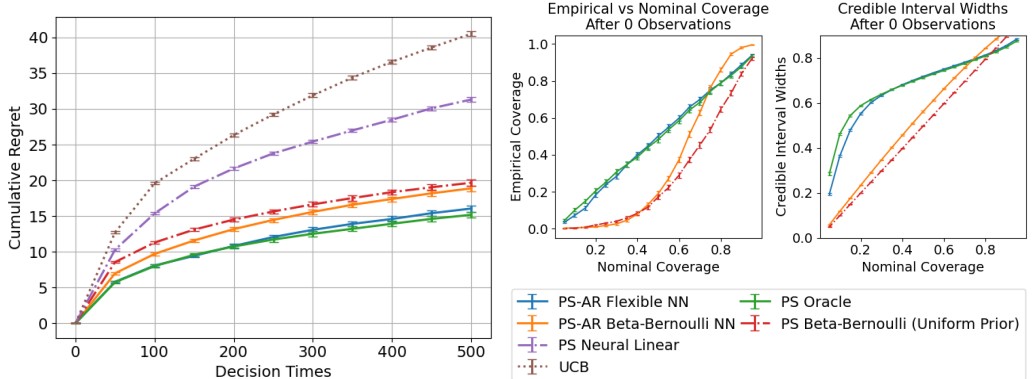

Figure 6: **Evaluation in mixture Beta-Bernoulli Setting.** Left: cumulative regret with $|\mathcal{A}^{\text{new}}| = 10$, averaged over 500 repetitions. Right: evaluating uncertainty quantification (coverage and interval width) averaged over 1000 actions not seen in training. Error bars are $\pm 1$ s.e.

# F  Additional Experiment Results

## F.1  Synthetic Setting: Mixture Beta-Bernoulli

Our synthetic experiments use a mixture model (Example 1) where $Z^{(a)} \in \mathbb{R}^2$ and the prior is a mixture of two Betas and the likelihood is Bernoulli. See Appendix G.2 for more details.

**Models.** We consider two sequence model $p_\theta$ variants. (i) FLEXIBLE NN is a neural network that takes $Z^{(a)}$ and a summary of the past outcomes for action $a$ as input. (ii) BETA-BERNOULLI NN, is the closed-form posterior predictive for the Beta-Bernoulli model from (12); its hyperparameters $\alpha_\theta(Z^{(a)})$ and $\beta_\theta(Z^{(a)})$ are parameterized by neural networks that take $Z^{(a)}$ as input.

**Regret: Figure 6 (Left).** PS ORACLE, which implements Thompson (posterior) sampling with a prior that matches the data generating process, has the lowest regret. PS-AR FLEXIBLE NN closely matches the performance of PS ORACLE. PS-AR BETA-BERNOULLI NN which uses a sequence model with a misspecified, unimodal Beta prior performs similarly to PS BETA-BERNOULLI (UNIFORM PRIOR) which performs exact Thompson sampling with a uniform prior. All these mentioned Thompson sampling-based algorithms outperform the UCB algorithm [1] and PS NEURAL LINEAR, Thompson sampling with a linear Gaussian bayesian model with an uninformative prior on top of learned text embeddings. See more on baseline algorithms in Appendix G.4.

**Uncertainty Quantification: Figure 6 (Right).** For 1000 actions $a$ not seen in training, we form 250 posterior samples $\hat{\mu}_1^{(a)}$ by autoregressively generating outcomes conditional on $Z^{(a)}$ using $p_\theta$. We use the percentiles of the sampled $\hat{\mu}_1^{(a)}$'s to form intervals and evaluate how often the true $\mu_1^{(a)}$ is within these intervals; ideally, an 80% interval contains $\mu_1^{(a)}$ 80% of the time. The intervals generated by the FLEXIBLE NN sequence model have excellent coverage; moreover, the width of the intervals are the narrowest that have correct coverage (matching PS ORACLE). In contrast, the BETA-BERNOULLI NN sequence model which has a unimodal (misspecified) Beta prior has worse coverage.

# G  Experiment Details

In this appendix we discuss general implementation techniques in Appendix G.1, synthetic experiments in Appendix G.2, news article recommendation experiments in Appendix G.3, and bandit algorithms in Appendix G.4.

## G.1  General Implementation Techniques

**(1) Bootstrapping Training Data.** The sequence length $n$ in the training set $\mathcal{D}^{\text{hist}} := \{Z^{(a)}, Y_{1:n}^{(a)} : a \in \mathcal{A}^{\text{hist}}\}$ may be smaller than the decision horizon $T$. To ensure the learned sequence model $p_\theta$ has

low prediction loss, $\ell_T(p_\theta)$, for longer sequences, we bootstrap the data in training by computing the loss with $\tilde{Y}_{1:T}^{(a)}$ where $\tilde{Y}_{1:T}^{(a)}$ are sampled with replacement from $Y_{1:n}^{(a)}$ (see Appendix A for details).

**(2) Truncating Generation Lengths.** When the population size $T$ is large, generating missing outcomes for the entire population can be costly. To save computation, we implement a slightly modified version of PS-AR that instead generates only $m$ missing outcomes per action and averages those $m$ outcomes to form $\hat{\mu}_t^{(a)}$; by (21) this is a good approximation when $m$ is relatively large. This is further supported by our simulation results where we vary $m$ (see Figure 5 in Appendix A).

### G.2    Synthetic Experiments: Mixture Beta-Bernoulli

**Data generating process**    In this setting, we use article attributes be $Z^{(a)} = \left(Z_1^{(a)}, Z_2^{(a)}\right) \in \mathbb{R}^2$ where $Z_1^{(a)}, Z_2^{(a)} \overset{i.i.d.}{\sim}$ Uniform$(0, 0.25)$. We sample $Y_{1:T}^{(a)}$ by first sampling $\mu_\infty^{(a)} \in [0,1]$ from a mixture:

$$\mu_\infty^{(a)} \mid Z^{(a)} \sim \begin{cases} \text{Beta}\left(25Z_1^{(a)} + 1, \ 25(1 - Z_1^{(a)}) + 1\right) & \text{w.p. } 1/2 \\ \text{Beta}\left(25(1 - Z_2^{(a)}) + 1, \ 25Z_2^{(a)} + 1\right) & \text{w.p. } 1/2 \end{cases}$$

Then, outcomes are sampled as $Y_1^{(a)}, \dots, Y_T^{(a)} \mid \mu_\infty^{(a)}, Z^{(a)} \overset{i.i.d.}{\sim}$ Bernoulli$(\mu_\infty^{(a)})$.

Here, $\mu_\infty^{(a)}$ corresponds to the success rate in the data generating process, in contrast to $\mu^{(a)}$ in the main text of the paper, which corresponds to the mean (or success rate) in the finite-sample population of size $T$. Note that $\mu^{(a)}$ converges to $\mu_\infty^{(a)}$ as $T$ goes to infinity.

**Training and Validation Datasets**    The training and validation datasets contain 2500 and 1000 articles each, respectively. During training (Algorithm 1), we use $T_{\text{train}} = 500$. Hyperparameters and early stopping epochs are chosen using the validation dataset.

**Additional model and training details**

- FLEXIBLE NN. This model implements the autoregressive model as a neural network that takes as input the action/article attribute $Z^{(a)}$ (a vector in $\mathbb{R}^2$) and summary statistics of observations for this action, and outputs a value (probability) in $[0, 1]$. The summary statistic we use is simple because outcomes $Y_t^{(a)}$ are binary; specifically it consistes of a tuple with the mean of outcomes from action $a$, and the reciprocal of 1 plus the total number of outcome observations for action $a$, i.e. $\left(\frac{1}{N^{(a)}} \sum_{t'=1}^{t-1} Y_{t'} \mathbb{1}_{A_{t'}=a}, \ \frac{1}{1+N^{(a)}}\right)$, where $N^{(a)} := \sum_{t'=1}^{t-1} \mathbb{1}_{A_{t'}=a}$. (In practice, we found that repeating the summary tuple input 10 improved performance, so the model took as input vectors in $\mathbb{R}^{22}$ which consisted of a 2-dimensional $Z^{(a)}$ and 10 copies of the sufficient statistic tuple). Note that this entire $p_\theta$ could alternatively be implemented as a transformer.

  The MLP we use has three linear layers, each of width 50. After the first and second linear layers, we apply a ReLU activation. After the last linear layer, we apply a sigmoid function, so that the output is in $(0, 1)$. The models are trained for 1000 epochs with learning rate 0.001, batch size 500, and weight decay 0.01 using the AdamW optimizer.

- BETA-BERNOULLI NN. This is a sequential model that is the (closed-form) posterior predictive for a Beta-Bernoulli. The prior parameters for the Beta distribution, $\alpha_\theta(Z^{(a)})$ and $\beta_\theta(Z^{(a)})$, are each parameterized by separate neural network MLP models that take in $Z^{(a)}$.

  The MLPs we use has three linear layers, each of width 50. After the first and second linear layers, we apply ReLU activations. After the last linear layer, we also apply a ReLU activation, so that the final output is in $[0, \infty)$. We initialize weights so that the bias term for both $\alpha_\theta(Z^{(a)})$ and $\beta_\theta(Z^{(a)})$ to 1, so that we avoid starting with Beta parameters of value 0, as Beta parameters need to be positive. The models are trained for 1000 epochs with learning rate 0.001, batch size 500, and weight decay 0.01 using the AdamW optimizer.

**Additional details on Figure 6**    In our uncertainty quantification plots Figure 6 (right), we evaluate over all 1000 actions in the validation set. We form 250 samples of $\hat{\mu}_1^{(a)}$ for each action in the

validation set using Algorithm 3 with $m = 500$. To generate posterior samples for BETA-BERNOULLI NN, we use the closed-form posterior (i.e., $m = \infty$).

In our regret plots Figure 6 (left), we run 500 runs. In each run we randomly choose $|\mathcal{A}^{\text{new}}| = 10$ actions randomly with replacement from the validation set, and all algorithms are evaluated on these same sampled actions in each run. Regret is calculated relative to $\mu_\infty^{(a)}$ from the data generating process.

### G.3 News Recommendation Experiment Details

**Additional data details** The training and validation datasets contain 9122 and 2280 distinct actions/articles each, respectively. During training, we use $T_{\text{train}} = 500$ As in Appendix G.2, hyperparameters and early stopping epochs are chosen using the validation dataset.

We now discuss the news data preprocessing process. This dataset is free to download for research purposes at https://msnews.github.io/. It is under a Microsoft Research License at https://github.com/msnews/MIND/blob/master/MSR%20License_Data.pdf, which we comply with. The terms of use are at https://www.microsoft.com/en-us/legal/terms-of-use.

Our preprocessing procedure is as follows:

1. Collect all articles from the MIND "large" dataset (training split only) [53].

2. Remove any article with fewer than 100 total impressions.

3. Normalize the success probabilities to be centered around 0.5 in a way that preserves the ranking of $\mu^{(a)}$. We do this transformation to speed up the learning procedure (since it requires more data to learn small true Bernoulli success probabilities accurately). We leave simulations without this transformation to future work.

   Our transformation procedures as follows: Let $\mu_0^{(1)}, \dots, \mu_0^{(|\mathcal{A}|)}$ be the original empirical success probabilities (average click rate). We use $\mathcal{A}$ to denote all articles in the MIND large dataset. The new success probabilities are defined as follows for each $a \in \mathcal{A}$:

$$\mu_\infty^{(a)} \leftarrow \begin{cases} \mu_0^{(a)} & \text{if } \mu_0^{(a)} \in \{0, 1\} \\ \text{logit}^{-1}\left(\text{logit}(\mu_0^{(a)}) - \bar{\mu}_0\right) & \text{otherwise} \end{cases}.$$

   Above, $\bar{\mu}_0 \triangleq \frac{1}{|\mathcal{A}|} \sum_{a' \in \mathcal{A}} \text{logit}(\mu_0^{(a')})$ and $\text{logit}(x) \triangleq \log \frac{x}{1-x}$. See Figure 7 for comparison of the success probabilities (click rates) before and after the transformation.

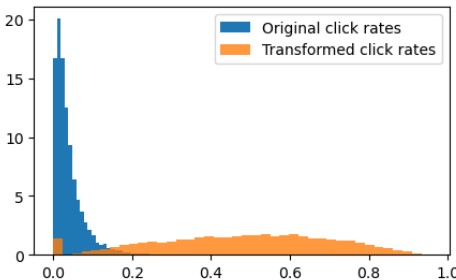

Figure 7: Original and transformed click rates. Note the spike at 0 for transformed click rates: only click rates that were not 0 or 1 are transformed.

4. Randomly select 20% of the remaining articles to be in the validation set; the rest are in the training set.

**Additional model details**

- FLEXIBLE NN (TEXT). This model very similar to the FLEXIBLE NN model in Appendix G.2, with the exception that in place of a two-dimensional $Z^{(a)}$, the MLP head of the neural network from before is fed as input a DistilBERT [46] embedding of text data $Z^{(a)}$.

Also, the MLP linear layers have width 100 instead of 50, and the sufficient statistics are repated 100 times instead of 10 times. All other architecture details are the same.

The model is trained for 500 epochs with learning rate 1e-5 on MLP heads, 1e-8 on the DistilBERT weights, batch size 500, and weight decay 0.01 using the AdamW optimizer.

- BETA-BERNOULLI NN (TEXT). This is very similar to the Beta-Bernoulli posterior predictive sequence model in Appendix G.2, with the exception that in place of a two-dimensional $Z^{(a)}$, the MLP head of the neural network from before is fed as input a DistilBERT [46] embedding of text data $Z^{(a)}$. On top of the one DistilBERT embedding are two separate MLP heads for $\alpha(Z^{(a)})$ and $\beta(Z^{(a)})$, which are trained together. Also, the MLP linear layers have width 100 instead of 50, and the sufficient statistics are repeated 100 times instead of 10 times. All other architecture details are the same.

  The model is trained for 500 epochs with learning rate 1e-5 on MLP heads, 1e-8 on the DistilBERT weights, batch size 500, and weight decay 0.01 using the AdamW optimizer.

- FLEXIBLE NN (CATEGORY). This is very similar to the flexible neural network model in Appendix G.2, but it uses a one-hot new category vector for $Z^{(a)}$ instead of a two-dimensional $Z^{(a)}$. The model architecture and training parameters are also the same.

- **DistilBERT.** Our two text models use DistilBERT [46] from https://huggingface.co/distilbert/distilbert-base-uncased. It has an apache-2.0 license, with license and terms of use at https://huggingface.co/datasets/choosealicense/licenses/blob/main/markdown/apache-2.0.md.

**Additional details on Figure 3**  In our uncertainty quantification plots Figure 3 (right), we evaluate over all 2280 articles/actions in the validation set. For our FLEXIBLE NN $p_\theta$ model, we form 250 samples of $\hat{\mu}_1^{(a)}$ for each action in the validation set using Algorithm 3 with $m = 500$. For our BETA-BERNOULLI NN $p_\theta$ model we use samples from the closed-form posterior.

In our regret plots Figure 3 (left), we run 500 runs. In each run we randomly choose $|\mathcal{A}^{\text{new}}| = 10$ actions randomly with replacement from the validation set, and all algorithms are evaluated on these same sampled actions in each run. Regret is calculated relative to $\mu_\infty^{(a)}$ as described above.

**Ensemble**  We describe the ensembling approach used in the uncertainty quantification plots in Figure 3 (right). To construct ensembles, we first train a DistilBERT model with an MLP head (MLP width 100, 3 layers, batch size 100, 500 epochs, learning rate 1e-5 on the head and 1e-8 on DistilBERT, weight decay 0.01, AdamW optimizer) to predict $Y_t^{(a)}$, using action/article features $Z^{(a)}$(headlines). Then, we freeze the DistilBERT weights, and train 50 MLP heads from scratch with random initialization and bootstrapped training data to create the ensemble (50 epochs, fixed DistilBERT embedding; other params the same as before).

### G.4  Bandit Algorithms

We compare our method with several baseline bandit methods.

**PS Beta Bernoulli (Uniform Prior)**  We model success rate $\mu_\infty^{(a)}$ and potential outcomes $Y_t^{(a)}$ using a conjgate Beta-Bernoulli model:

$$\mu_\infty^{(a)} \sim \text{Beta}(\alpha, \beta) \tag{18}$$

$$Y_1^{(a)}, \dots, Y_T^{(a)} \mid \mu_\infty^{(a)} \overset{i.i.d.}{\sim} \text{Bernoulli}(\mu^{(a)}) \tag{19}$$

We also use reward mappings $R(Y_t^{(a)}) := Y_t^{(a)}$. In our experiments, we use Beta-Bernoulli with a uniform prior, so $\alpha = \beta = 1$. Note that unlike the BETA-BERNOULLI NN, the prior here does not depend on action attributes $Z^{(a)}$.

Online decision-making uses Thompson sampling, as described in in Algorithm 6.

---

Algorithm 6: Beta-Bernoulli Posterior (Thompson) Sampling

---

1: **Inputs:** Prior hyperparameters $\alpha, \beta$.
2: Set priors $(\alpha_0^{(a)}, \beta_0^{(a)}) \leftarrow (\alpha, \beta)$, $\forall a \in \mathcal{A}^{\text{new}}$
3: **for** $t = 1, \ldots, T$ **do**
4:     **for** $a \in \mathcal{A}^{\text{new}}$ **do**
5:         Sample $\hat{\mu}^{(a)} \sim \text{Beta}(\alpha_{t-1}^{(a)}, \beta_{t-1}^{(a)})$
6:     **end for**
7:     Select action $A_t \leftarrow \arg\max_{a \in \mathcal{A}^{\text{new}}} \{\hat{\mu}^{(a)}\}$
8:     Observe outcome $Y_t$ from action $A_t$.
9:     Update posterior $(\alpha_t^{(A_t)}, \beta_t^{(A_t)}) \leftarrow \begin{cases} (\alpha_{t-1}^{(a)} + \mathbb{1}_{Y_t=1}, \beta_{t-1}^{(a)} + \mathbb{1}_{Y_t=0}) & \text{if } A_t = a \\ (\alpha_{t-1}^{(a)}, \beta_{t-1}^{(a)}) & \text{otherwise} \end{cases}$
10: **end for**

---

**PS Neural Linear**  We implement a very simple variant of "neural linear" as in Riquelme et al. [43], Snoek et al. [48]. Here, we model each arm reward as a Gaussian-Gaussian model. We fit the prior mean using item features $Z^{(a)}$, but set a shared prior variance across articles. Specifically,

$$\mu^{(a)} \sim N\left(g(Z^{(a)}), \sigma^2\right)$$
$$R_1^{(a)}, \ldots, R_T^{(a)} \overset{i.i.d.}{\sim} N\left(\mu^{(a)}, s^2\right)$$

First we address the choice of $g, \sigma^2, s^2$, which are chosen during pre-training, and then the bandit evaluation, which is standard Thompson sampling with a Gaussian-Gaussian model. We address these one at a time.

**Parameters**

1. First, $g$ is obtained by training a model to predict $\mu^{(a)}$ from just $Z^{(a)}$ (no history of past rewards), using the training set. For synthetic experiments, for $g$, we used a neural network with almost the same architecture as for the autoregressive model we use for this dataset. However, the model only takes $Z^{(a)}$ (and not previous rewards for $a$). We use all of the same hyperparameters as we did to train the autoregressive model for this dataset. For news datasets, we trained a DistilBERT model with a MLP on top that takes the embedded article headlines $Z^{(a)}$ as input (and not previous rewards for $a$). We use the same hyperparameters as we did to train the autoregressive model for this dataset, except for learning rates, which were chosen to be the best out of several (for synthetic, 1e-2; for news setting, 1e-5 for both the MLP head and the DistilBERT weights).

2. $\sigma^2, s^2$ were chosen to be reasonable values, which in our experiments were 0.25 for $s^2$ (which corresponds to maximum variance of a Bernoulli), and 1 for $\sigma^2$.

**UCB**  For UCB we use the multi-arm bandit algorithm described in Section 6 of [1]. We set the failure probability $\delta = 0.1$ and use sub-Gaussian parameter 0.5 (since we have binary rewards).

**SquareCB**  In these experiments, we use the flexible neural network $p_\theta$ with text attributes, but instead of using Thompson sampling, we use SquareCB [18], which is a bandit algorithm that uses a regression oracle to predict the value of each action. Note that our setting differs from the setting of SquareCB [18], as SquareCB assumes that the prediction model for action value is being learned online, while our prediction model has been pretrained on historical data and is not learned online.

For setting the learning rate $\gamma$ in SquareCB [18], we follow Foster et al. [19] and consider a time-varying learning rate $\gamma_t = \gamma_0 t^\rho$, where $\gamma_0 \in \{10, 100\}$ (a subset of those suggested in Foster et al. [19]), and $\rho \in \{0.25, 0.5\}$ are hyperparameters. While some hyperparameter combinations for SquareCB perform almost as well PS-AR FLEXIBLE NN (TEXT), we remark that there is no principled approach to choosing the learning rate provided in existing works (besides grid search by deploying the algorithm many times).

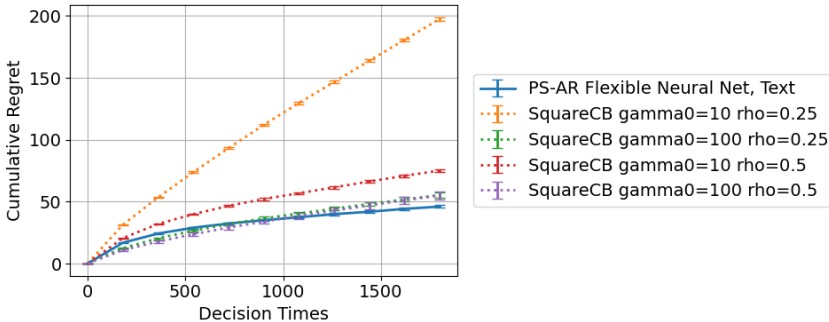

Figure 8: Regret comparison on news dataset for SquareCB and posterior sampling, both using the flexible neural network sequence model using text attributes in 3.2.

### G.5 Additional Synthetic Experiments: Recovering the True Prior / Empirical Bayes

Here, we demonstrate in practice an setting where we perform "empirical Bayes" using our pretraining procedure (Algorithm 1). We find that we recover the true prior fairly well.

**Data generation** We use a synthetic Beta-Binomial data generating process. We consider one-dimensional action features $Z^{(a)} \overset{i.i.d.}{\sim}$ Uniform$(0, 1)$. We then sample $\mu^{(a)}$ from a Beta distribution, where

$$\mu^{(a)} \mid Z^{(a)} \sim \text{Beta}\big(Z^{(a)} \cdot 5 + 1, \ (1 - Z^{(a)}) \cdot 5 + 1\big). \tag{20}$$

Then, $Y^{(a)} \mid \mu^{(a)}, Z^{(a)} \sim$ Bernoulli$(\mu^{(a)})$. We use $R(y) := y$. We use a training dataset of size 25,000 actions and a validation set of size 10,000 actions; both datasets have observation sequences of length $n = 500$.

**Autoregressive model** We use $p_\theta$ which matches the posterior predictive of a Beta-Bernoulli distribution, akin to (12). To accomodate $Z^{(a)}$ features, we parameterize the prior hyperparameters: $\alpha_\theta(Z^{(a)}), \beta_\theta(Z^{(a)})$ (we follow the procedure described in Appendix G.2 for BETA-BERNOULLI NN). The neural network model architecture used in $\alpha_\theta(Z^{(a)}), \beta_\theta(Z^{(a)})$ and the training procedure are also the same as described for BETA-BERNOULLI NN in Appendix G.2 (except that the MLP widths are 100).

**Recovering the Prior: Figure 9** We show in Figure 9 that through our pretraining procedure Algorithm 1 with our particular choice of $p_\theta$ model class, that we (approximately) recover the true prior. We show this by comparing means and standard deviations of samples from our learned prior (using $p_\theta$) vs. the true prior (according to the data generating process), for different draws of $Z^{(a)}$. In the scatter plots, each point corresponds to one $Z^{(a)}$.

Specifically, in these plots we use 100 actions sampled uniformly from the validation set. For each of these 100 actions we form $10,000$ samples of $\hat{\mu}_1^{(a)}$ using Algorithm 3 using our learned $p_\theta$ model. We also form $10,000$ samples from the true data generating prior (20) for each of the 100 actions. Then for each action, we compute the mean and standard deviations of the samples on the "prior" samples $\hat{\mu}_1^{(a)}$ from $p_\theta$; we also compute the mean and standard deviations of the samples from the true prior. We then plot these in a scatter plot; for each action, we have the prior mean according to $p_\theta$ vs the prior mean according to the data generating process—this forms one point on the scatter plot. A similar procedure is plotted on the right. There, instead of computing the mean of the prior samples, we compute a measure of the spread of the prior samples: let $\hat{\mu}_{1,1}^{(a)}, \hat{\mu}_{1,2}^{(a)} \dots, \hat{\mu}_{1,10000}^{(a)}$ be the prior samples. Let $\bar{\mu}_1^{(a)} = \frac{1}{10000} \sum_{i=1}^{10000} \hat{\mu}_{1,i}^{(a)}$. Then we compute the mean absolute deviation $\frac{1}{10000} \sum_{i=1}^{10000} \big|\hat{\mu}_{1,i}^{(a)} - \bar{\mu}_1^{(a)}\big|$ for this set of prior samples.

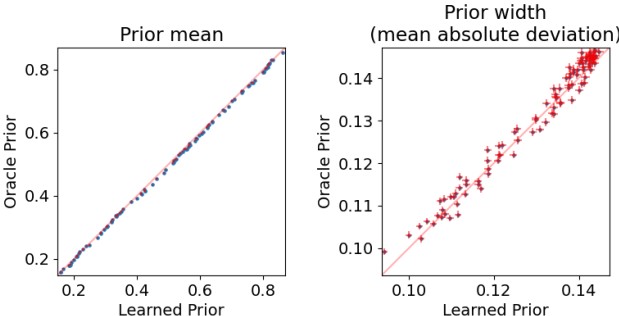

Figure 9: **Comparing oracle prior vs prior learned through our method (empirical bayes) in a synthetic setting**. Error bars represent $\pm 1$ standard error; the error bars on the left plot are present but small enough to not be visible.

## G.6 Compute resources

Unless otherwise specified, computational experiments were run on an internal CPU cluster at Columbia GSB, where for any individual run we request at most 50GB of memory, and each individual training run takes at most an hour or two.

We train on a GPU only for text models that fine-tune DistilBERT, which applies to learning sequence models for the MIND news setting (Section 3.2, Appendix G.3). In these cases, each run uses a single NVIDIA A40 GPU, and a single training run of the slowest variant FLEXIBLE NN (TEXT) takes about 16 hours to complete. We ran hyperparameter sweeps and tried several architecture variations, so we estimate the total compute to be an order of magnitude or two larger than that of a single run. Note that we use the CPU cluster for e.g. ensembling, where we freeze the DistilBERT part of the model and train MLP heads with random initializations on bootstrap-resampled subsets of the data.

# H   Related Work

**Meta-Learning in Bandits.** There are a variety of bandit algorithms for meta-learning problems [51, 8, 26, 3]; these methods primarily focus on simpler settings (e.g. Gaussian or linear reward models). There are also deep meta-learning methods developed for recommendation systems and the cold-start problem [52, 56, 57]. These works primarily focus on more complex recommendation settings (e.g. tracking the same user over time) and not on uncertainty. In contrast, our goal is to showcase our uncertainty quantification method for decision making in a semi-realistic setting.

**Reinforcement Learning (RL) with Pre-Trained Autoregressive Models.** Many recent works in RL leverage sequence models that are pretrained on a large volume of data collected by an expert policy. [29, 30] relate sampling from a model that predicts the next expert action to Thompson sampling. Other works apply goal-conditioned sampling of expert actions to improve over average expert behavior [54, 24, 9, 12, 13]; this works well in some settings but is provably sub-optimal others [4, 32]. Our work is different: we use sequence models to imagine plausible trajectories of future rewards, and use this to drive intelligent decision-making without requiring expert demonstrations.

**Thompson Sampling with Deep Learning Models.** Several classes of approaches that have emerged to scale Thompson sampling to modern large scale decision-making problems that utilize neural network. The first class places a Bayesian prior on the weights of the neural network itself. These methods include those that form a Bayesian linear regression model from the last layer of a trained neural network [43, 48], as well as Bayesian neural networks [55]. A second class of approaches involves forming using an ensemble of neural networks to simulate samples from a posterior distribution [37, 31, 41]. This class also includes algorithms that build on Epinets [40, 58, 39], which attempt to retain the performance of the ensembling with lower computational cost. Notably, [40] uses sequence prediction loss to *evaluate* the quality of ("epistemic") uncertainty quantification, inspiring our efforts to construct bandit algorithms using sequence models.

# I Discussion

We formulate a loss minimization problem that implicitly learns an informed prior using historical data, in order to model the posterior distribution of action rewards for decision-making. This connection enables using modern ML tools to learn rich representations to comprehend uncertainty, in an actionable way. Our formulation introduces a fresh approach to the longstanding challenge of scaling Thompson sampling to incorporate neural networks that incorporate unstructured inputs such as images and text [43]. The main ideas behind our algorithm generalize to *contextual* settings where user-specific contexts $X_t$ can be used to tailor recommendation decisions. We describe generalizing our method to this setting in Appendix D and leave a deeper dive to future work.

**Limitations.** We assume articles are i.i.d. between pretraining and online evaluation, and user outcomes for each action are exchangeable. Such assumptions may not be appropriate in practice, e.g., if user preferences are nonstationary. In conducting this work, we struggled to find publicly available datasets on which to evaluate our method, which led us to build our news recommendation setting. Building public benchmarks for bandit problems that require using complex inputs (e.g. text and/or images) for best performance is an important open direction. A limitation of this work is we do not provide a thorough answer as to the quality of the historical data (e.g., amount of data and/or how data was collected) necessary to ensure learning good sequence models.

# J Additional exchangeability comment

We elaborate on a comment made in the text. Exchangeability means that outcomes from recommendations made to a large subset of $m < T$ users is likely to be representative of outcomes that would have been observed among all $T$ users.

It is not hard to formalize results of this type. For instance, for any permutation $\sigma$ over $\{1, \ldots, T\}$,

$$\left( \mathbb{E}\left[ \left( \frac{1}{m} \sum_{i=1}^{m} R(Y_{\sigma(i)}^{(a)}) - \frac{1}{T} \sum_{t=1}^{T} R(Y_t^{(a)}) \right)^2 \right] \right)^{1/2} \leq \sqrt{\frac{1/4}{m}} \times \sqrt{1 - \frac{m}{T}} \qquad (21)$$

The term $\sqrt{1 - \frac{m}{T}}$ is the finite population correction to the standard error of the sample mean [42, Ch 4.5]

# K When is an Autoregressive Sequence Model a Valid Posterior Predictive?

In Algorithm 1, we learn an autoregressive model to use in place of a posterior predictive in Algorithm 2. We make this connection in Section 3.1 and establish a regret bound for Algorithm 2 that holds whenever $p_\theta$ has low loss.

In this section, we address the following question: *When is $p_\theta$ a valid posterior predictive, for some underlying Bayesian model?*

In order for an autoregressive generative sequence model to be a valid posterior predictive distribution, the sequence model to be *infinitely exchangeable*. We say that a sequence model is an infinitely exchangeable sequence model if it generates infinitely exchangeable random variables (Definition 2).

**Definition 2** (Exchangeablity). *A sequence of random variables $Y_1, Y_2, \ldots, Y_n$ is exchangeable if for any permutation $\pi$, the following are equal in distribution:*

$$\left( Y_1, Y_2, \ldots, Y_n \right) \overset{D}{=} \left( Y_{\pi(1)}, Y_{\pi(2)}, \ldots, Y_{\pi(n)} \right).$$

*An infinite sequence of random variables is infinitely exchangeable if any finite subset is exchangeable.*

Practically, this means that the models we train need to be invariant to the *order* in which previous outcomes are fed into the model. The key insight behind why infinitely exchangeable sequence models are valid posterior predictives is De Finetti's Representation Theorem (Theorem 2 below). We state this Theorem for binary outcomes for simplicity [10, 22], but it generalizes to real-valued outcomes [11].

**Theorem 2** (De Finetti's Representation Theorem for Binary Outcomes)**.** *If a sequence of binary random variables $\{Y_i\}_{i \in \mathbb{N}}$ is infinitely exchangeable, then there exists a unique distribution $P(\mu)$ on*

$[0, 1]$ *such that for some* $\mu \sim P(\mu \in \cdot)$,

$$Y_1, Y_2, Y_3, \cdots \mid \mu \overset{i.i.d.}{\sim} \text{Bernoulli}(\mu).$$

The implication of Theorem 2 is that *any* infinitely exchangeable sequence of binary random variables $\{Y_i\}_{i \in \mathbb{N}}$ can equivalently be described as being generated by a particular Bayesian model with a Bernoulli likelihood. Above, $\mu$ is a latent success probability that is drawn from some prior distribution $P(\mu \in \cdot)$.

In practice note that our bootstrap training procedure in Algorithm 1 helps ensure our sequence model $p_\theta$ is approximately exchangeable.

## L   Broader Impacts

Although our contribution is largely conceptual/theoretical, we list some potential positive and negative societal impacts.

**Positive impacts**   Our method could potentially enable better decision making in a variety of settings, including areas of clear social good like personalized healthcare. By incorporating historical data, including with complex features (e.g. text or images), in a flexible way, we enable more historical data to be used more effectively.

**Negative impacts**   Our method proposes learning recommendation algorithms using historical data. It is possible that historical data contains biases that can be harmful when perpetuated. We urge researchers to be thoughtful when curating and using historical data for this purpose. We also urge researchers to be thoughtful about the rewards $R(\cdot)$ they are aiming to maximize, as e.g. it is not always best for the user to maximize the user's time spent on a recommendation platform.

