# OpenReview forum: "Posterior Sampling via Autoregressive Generation"
_NeurIPS.cc/2024/Workshop/BDU — NeurIPS BDU Workshop 2024 Poster_

### Official Review · Reviewer_ywdW · 2024-09-17

**Rating:** 6
**Confidence:** 3

**Review:**

**Overview:**
The paper presents a novel framework for learning bandit algorithms from historical data, specifically applied to a cold-start recommendation problem. The authors propose using an autoregressive model to predict sequences of feedback/rewards, which implicitly learns an informed prior. This approach is framed as an implementation of Thompson Sampling with a learned prior. The paper provides theoretical guarantees and demonstrates the framework on a news recommendation task.

**Strength:**
The use of autoregressive models for posterior sampling in bandit problems is a novel approach. There are solid theoretical results including a novel regret bound that scales with the pre-training loss of the sequence model, which highlights the connection between pre-training loss and online decision-making performance. Emperical results seem strong, outperforming other approaches in terms of cumulative regret.

**Weekness:**
Comparison to other state-of-the-art methods and the tasks chosen for expeirments are limited. The performance will be more convincing if comparison with e.g., "Learning to Optimize Via Posterior Sampling, Daniel Russo" and "TS-UCB: Improving on Thompson Sampling, Jackie Baek" are made. These important citations are also missing. Experiments on other standard bandit/decision making datasets should also be beneficial. Finally, the complexity of training and deploying autoregressive models might be a concern for large-scale applications.

Nonetheless, for a workshop paper, the idea it presents is interesting enough.

---

### Official Review · Reviewer_iNFk · 2024-09-29
**Review for "Posterior Sampling via Autoregressive Generation"**

**Rating:** 7
**Confidence:** 5

**Review:**

1. Summary of the Paper:
This paper presents a novel approach to bandit algorithms, particularly focusing on Thompson sampling, by leveraging autoregressive generative models. The authors propose a framework that uses historical data to pretrain an autoregressive model for predicting sequences of rewards. This model is then used to implement Thompson sampling by generating imagined reward sequences for each action. The work is applied to a cold-start recommendation problem, specifically in news article recommendation.

2. Strengths:
a) Novelty: The integration of autoregressive generative models with Thompson sampling is an innovative approach to bandit algorithms. This bridges the gap between modern machine learning techniques and classical decision-making strategies.

b) Theoretical Grounding: The authors provide a theoretical analysis linking their pretraining loss to online decision-making performance, which adds rigor to their approach.

c) Practical Application: The framework is demonstrated on a real-world problem (news recommendation), showing its potential for practical impact.

d) Scalability: The method's ability to leverage large historical datasets for pretraining suggests good scalability to large-scale problems.

3. Weaknesses:
a) Empirical Evaluation: The paper would benefit from a more comprehensive empirical evaluation, including comparisons with a wider range of baseline methods and experiments on diverse datasets.

b) Computational Complexity: There's limited discussion on the computational demands of the approach, particularly for large-scale applications. This is crucial for understanding its practical feasibility.

c) Exchangeability Assumption: The assumption of exchangeability might limit the applicability of the method in some real-world scenarios where context significantly changes over time.

4. Detailed Comments:
- The theoretical analysis (Theorem 1 and Corollary 1) is a strong point, but it would be helpful to provide more intuitive explanations of these results for readers less familiar with the technical details.
- The integration of language models for processing news article headlines is interesting, but more details on this process and its impact would be valuable.
- The paper could benefit from a more in-depth discussion of the limitations of the approach and potential future work to address these limitations.
- A comparison with recent work on approximate Thompson sampling and hypermodels would strengthen the paper's positioning in the current literature.

5. Questions for Authors:
1. How does the computational complexity of your approach compare to traditional Thompson sampling methods, especially for large action spaces?
2. Can you elaborate on how your method might be extended to non-exchangeable settings, such as those with evolving user preferences over time?
3. Have you explored the impact of different architectures for the autoregressive model on the performance of your method?
4. How sensitive is your approach to the quality and quantity of historical data available for pretraining?

6. Recommendation:
This paper presents a novel and potentially impactful approach to bandit algorithms, combining modern machine learning techniques with classical decision-making strategies. While there are areas for improvement, particularly in empirical evaluation and addressing limitations, the theoretical contributions and practical applicability make this work valuable to the field.

The authors should focus on:
1. Expanding the empirical evaluation with more diverse baselines and datasets.
2. Providing a more detailed discussion of computational complexity and scalability.
3. Addressing the limitations of the exchangeability assumption and discussing potential extensions.
4. Enhancing the comparison with recent related work in approximate Thompson sampling and hypermodels/hyperagent.

With these revisions, this paper could make a significant contribution to the field of online decision-making and recommendation systems.

---

### Decision · Program_Chairs · 2024-10-09

Accept (Poster)